# Structural basis for malate-driven, pore lipid-regulated activation of the Arabidopsis vacuolar anion channel ALMT9

Yeongmok Lee [1,8], Elsa Demes-Causse[2,8], Jaemin Yoo [3], Seo Young Jang [4], Seoyeon Jung [1], Justyna Jaślan [2], Geum-Sook Hwang [4,5], Jejoong Yoo[3], Alexis De Angeli [2] & Sangho Lee [1,6,7] ✉

In plant cells, ALMTs are key plasma and vacuolar membrane-localized anion channels regulating plant responses to the environment. Vacuolar ALMTs control anion accumulation in plant cells and, in guard cells, they regulate stomata aperture. The activation of vacuolar ALMTs depends on voltage and cytosolic malate, but the underlying molecular mechanisms remain elusive. Here we report the cryo-EM structures of ALMT9 from *Arabidopsis thaliana* (AtALMT9), a malate-activated vacuolar anion channel, in plugged and unplugged lipid-bound states. In all these states, membrane lipids interact with the ion conduction pathway of AtALMT9. We identify two unplugged states presenting two distinct pore width profiles. Combining structural and functional analysis we identified conserved residues involved in ion conduction and in the pore lipid interaction. Molecular dynamics simulations revealed a peculiar anion conduction mechanism in AtALMT9. We propose a voltage-dependent activation mechanism based on the competition between pore lipids and malate at the cytosolic entrance of the channel.

The regulation of stomata aperture is crucial for the adaptation of plants to terrestrial environment. Indeed, stomata opening controls water and gas exchange between leaves and atmosphere, thereby playing a pivotal role in plant responses to stresses. Opening and closure of the stomata depends on the swelling and shrinking of the two guard cells delimiting the stomata pore[1]. Vacuolar volume changes occurring in the guard cells are driven by the modification of the vacuole that can increase its volume up to 40 %[2,3]. Such modifications of the vacuolar volume rely on the transport of osmolytes across the vacuolar membrane to accumulate/release solutes to modify the osmotic pressure of the guard cell[1]. Among the different ion transporters and ion channels localized in the vacuolar membrane, the ion channels of the aluminum-activated malate transporter (ALMT) family

play a key role for the transport of chloride and malate during stomata opening/closure[4]. The ALMT family is specific to plants and is named based on the function of the first identified member in mediating aluminum-activated malate efflux in the roots of *Triticum aestivum*[5]. However, not all ALMTs are activated by aluminum. ALMTs are also involved in transporting various anion including chloride, nitrate, fumarate, tartrate, citrate, and malate[5–13]. ALMTs have evolved into more than five different clades[14,15] depending on their functional niche: Aluminum resistance[5,16], mineral acquistion[10,17,18], pollen tube growth[19], fruit acidity[11], stomatal opening[9], and closure[20]. In the model plant *Arabidopsis thaliana*, 14 members of the ALMT family are divided into different clades with different functional properties[14].

[1]Department of Biological Sciences, Sungkyunkwan University, Suwon, Republic of Korea. [2]IPSiM, CNRS, INRAE, Institut Agro, Université Montpellier, Montpellier, France. [3]Department of Physics, Sungkyunkwan University, Suwon, Republic of Korea. [4]Integrated Metabolomics Research Group, Metropolitan Seoul Center, Korea Basic Science Institute, Seoul, Republic of Korea. [5]College of Pharmacy, Chung-Ang University, Seoul, Republic of Korea. [6]Biomedical Institute for Convergence at SKKU, Sungkyunkwan University, Suwon, Republic of Korea. [7]Department of Metabiohealth, Sungkyunkwan University, Suwon, Republic of Korea. [8]These authors contributed equally: Yeongmok Lee, Elsa Demes-Causse. ✉e-mail: sangholee@skku.edu

It is emerging that ALMTs feature unique folds and electrophysiological properties among membrane proteins[8,21]. The structures of ALMT1 from *A. thaliana* in clade 1 (AtALMT1), an aluminum-activated malate transporter, reveal aluminum-induced reorganization of two loops connecting transmembrane helices (TMs), leading to the opening of the transporter[21]. ALMT12/QUAC1 from *A. thaliana* in clade 3 is known as an anion channel regulating stomatal closure[20]. The structure of an orthologous ALMT12/QUAC1 from *Glycine max* (GmALMT12) presents a twisted two-layer dimeric interface by transmembrane and cytoplasmic domains[8]. Phosphorylation of the intracellular domain (ICD) modulates a conformational switch of the transmembrane domain (TMD) for channel opening of GmALMT12[8,22]. Unlike AtALMT1, GmALMT12 has lateral fenestrations filled by densities that could not be modeled due to the limited resolution (3.5 Å). AtALMT1 shows aluminum-induced activation within minutes and mediates voltage-dependent malate efflux conferring aluminum tolerance[16]. By contrast, GmALMT12 exhibits voltage-dependent and rapid activation kinetics similar to those of AtALMT12[8,20]. Differently from the plasma membrane ALMTs such as AtALMT1 and AtALMT12, the vacuolar ALMTs (i.e., AtALMT4, AtALMT6, and AtALMT9) present voltage-dependent and slow (in seconds) activation kinetics with different selectivity and regulation characteristics[7,9,23]. The distinctive slow activation kinetics of the vacuolar ALMTs suggest that uncharted modulatory elements may contribute to the activation of the ion transport activity.

The evolution of a wide range of ion channels and transporters modulates the exchanges of various inorganic ions and organic molecules across the cellular membranes[24]. Notably, the lipids in these membranes have strongly influenced the evolution of ion channels. Recent advances in cryo-EM structure determination uncovered the intricate relationship between membrane lipids and ion channels[25]. Phospholipids and sterols allosterically modulate the activity of various ion channels including voltage-gated channels[26–28], ligand-gated channels[29–34], and mechanosensitive channels[35–39]. Furthermore, direct modulations of ion channels by the recruitment of pore lipids are observed in mechanosensitive channels such as OSCA[40], TMC-1[41], TREK1[42], and MscS[43]. However, the recruitment of lipids in the pore of non-mechanosensitive channels remains elusive.

To unveil an activation mechanism and modulatory elements underlying slow activation kinetics at molecular level in the clade 2 ALMTs, we report structures of ALMT9 from *Arabidopsis thaliana* (AtALMT9) in nine lipid-bound states: two sterol mimic-bound classes in plugged states, five pore lipid-bound classes in plugged states, and two unplugged classes in a narrow and a wide pore. AtALMT9 presents pore and lateral fenestrations occupied by lipids and sterol mimics. Although AtALMT9 is not a mechanosensitive ion channel, lipids directly influence the shape of the permeation pore as well as its accessibility to solvents. Based on structural, molecular dynamics simulation, mass spectrometry, and electrophysiological data, we propose an AtALMT9 activation mechanism where malate and membrane potential induce conformational changes and lipid movements linked to the conductive state.

## Results

### Diverse lipid-bound states of AtALMT9

AtALMT9 presents an electrophysiological current evoked by membrane potential changes consisting of three components: a first instantaneous activation followed by a time-dependent current increase and, in presence of chloride, current fluctuations[9] (Supplementary Fig. 1). To elucidate structural basis for these electrophysiological properties, we purified AtALMT9-superfolder GFP (sfGFP)-hemagglutinin (HA) tag-decahistidine ($H_{10}$) tag with lauryl maltose neopentyl glycol (LMNG) detergent and collected three cryo-EM datasets from different conditions depending on the presence of malate and cholesteryl hemisuccinate (CHS), a sterol mimic

(Supplementary Fig. 2, Supplementary Table 1). Dataset 1 was collected in the presence of LMNG and CHS; dataset 2 in the presence of LMNG, CHS and malate; and dataset 3 in the presence of LMNG and malate (Supplementary Fig. 2). In the three datasets, all the observed structural classes of AtALMT9 exist as dimers and exhibit topologies similar to those of other ALMT structures[8,21]: a short N-terminal helix (αN) followed by six transmembrane helices (TM1–6) and six helices (H1–6) of ICD (Fig. 1a, b). In a dimer, the pore is formed by four transmembrane helices: two TM2s and two TM5s (Fig. 1a, c, d). Lateral fenestrations, previously observed in GmALMT12 structure[8], are also found in the AtALMT9 structures (Fig. 1a–d). Unlike a single fenestration in each protomer of GmALMT12[8], each AtALMT9 protomer harbors two lateral fenestrations ($F_{A1}$ and $F_{A2}$ in the promoter A, and $F_{B1}$ and $F_{B2}$ in the protomer B of a dimer) that are connected to the ion conduction pore found at the center of a dimer (Fig. 1c–e). The two lateral fenestrations of each protomer are delimited by TM2 and 5 (hereinafter called as "pore helices"), and TM3 and 6 (hereinafter called as "fenestration helices") (Fig. 1c, d). We observed heterogeneous densities across the pore and the four lateral fenestrations, reminiscent of the unidentified densities in GmALMT12 because of the resolution (3.5 Å) of the map[8] (Fig. 1c, d). However, thanks to the high-resolution consensus maps in datasets 2 (2.9 Å) and 3 (2.5 Å) we could determine that these densities across the pore and the four lateral fenestrations correspond to lipids.

We classified these heterogeneous densities into separated classes according to types of the bound lipids (Fig. 1f, Supplementary Figs. 3–7). From the CHS-supplemented dataset 1 with no malate and dataset 2 with 10 mM malate, we unequivocally identified CHS densities in the lateral fenestrations and phospholipid (PL) densities in the pore (hereinafter called as "pore lipid") in two classes (Fig. 1f, Supplementary Figs. 3, 4, 7). The only difference observed between the datasets 1 and 2 was the presence of a malate density in the pore of the dataset 2 (Fig. 1f, Supplementary Fig. 7). Given its lower resolution and its similarity with the dataset 2, we did not model the dataset 1. Unexpectedly, we observed that the succinate head group of CHS interacted with several residues. Since the succinate group was the key structural moiety to differentiate CHS from native cholesterols, these interactions were likely to present artificial features and to inhibit pore lipid expulsion (described later). Subsequently, we collected the CHS-free dataset 3 with 10 mM malate and observed extended densities in the lateral fenestrations in several classes (Fig. 1f, Supplementary Fig. 7). These densities were not compatible with CHS or LMNG both harboring two short tails (10 carbons) (Supplementary Fig. 7). These extended densities were likely to result from an acyl chain of a PL in the periphery of TMD (hereinafter called as "peripheral lipid") (Fig. 1f, Supplementary Fig. 7). To confirm the presence of PLs, we extracted lipids from the sample purified with the dataset 3 condition based on the Folch's method[44]. The lipid extract was analyzed by liquid chromatography with tandem mass spectrometry (LC-MS/MS). We detected various AtALMT9-bound PLs in the following descending order of abundance: phosphatidylethanolamine (PE), phosphatidylserine (PS), phosphatidylcholine (PC), phosphatidylinositol (PI), and phosphatidylglycerol (PG), but no detectable phosphatidic acid (PA) (Fig. 1g–i, Supplementary Fig. 8). In comparison with reported lipid compositions of Sf9 insect cells[45], anionic PS or acyl chain with 18 carbon atoms were slightly favored for AtALMT9 binding, but there was no significant preference for a specific lipid (Fig. 1g, h). These data were consistent with our structural observations showing lipid chains with 16 to 18 carbon atoms (Fig. 1h, Supplementary Figs. 7, 8).

We classified the nine structural classes of AtALMT9 in three groups based on the type of lipids and on their positions in the structure: sterol mimic-bound, pore lipid-bound, and peripheral lipid-only (Fig. 1f). In the CHS-supplemented datasets 1 and 2, we identified two distinct classes in the sterol mimic-bound state: "sterol 1" when one pore lipid tail of the sterol mimic (CHS) is in the fenestrations and

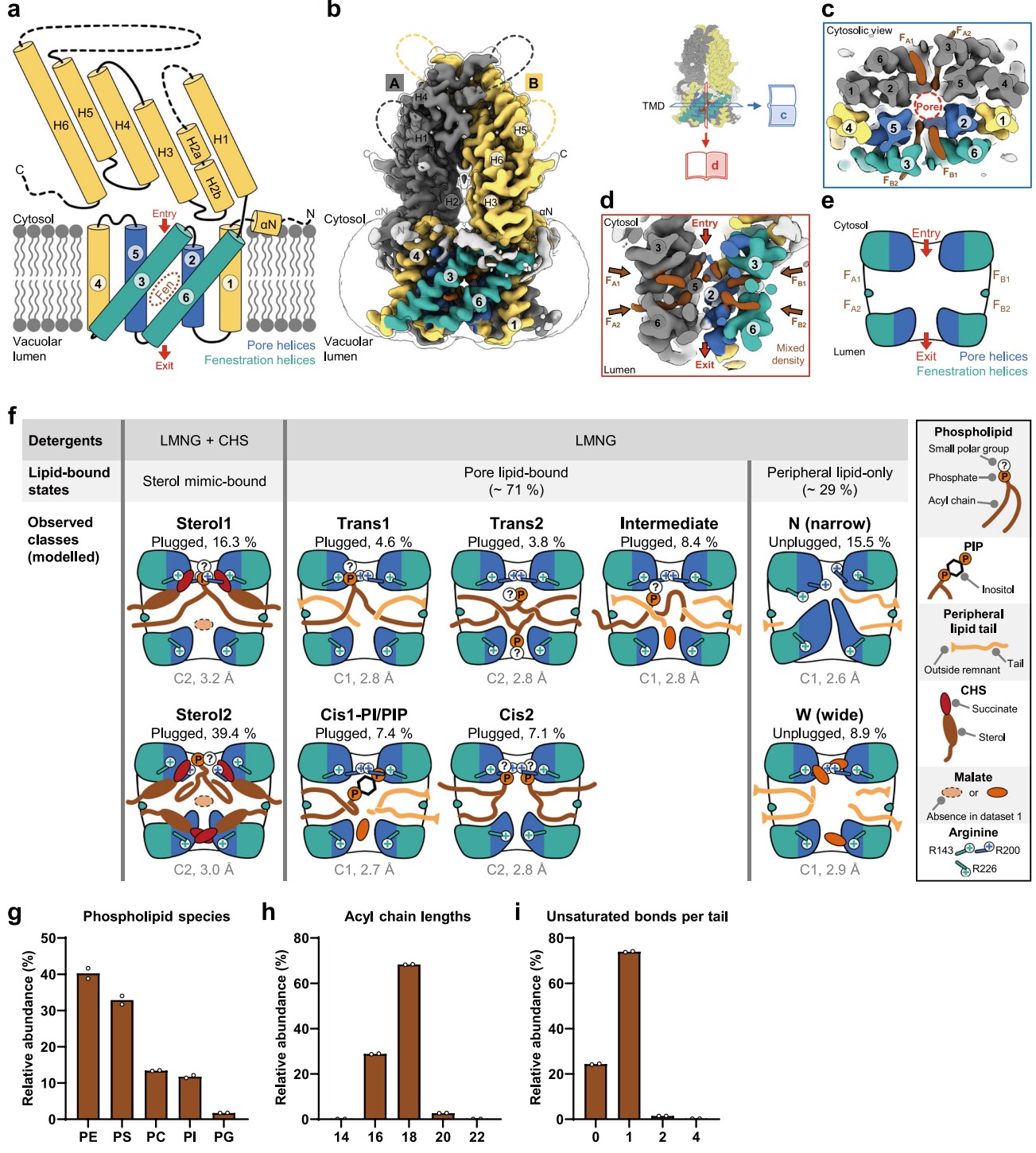

**Fig. 1 | Conformational landscape of AtALMT9 according to lipid-bound state.**
**a** Overall topology of AtALMT9. N- and C-termini are shown. An N-terminal helix is indicated with αN; transmembrane helices are indicated with corresponding numbers in circles. Pore helices are colored as blue, fenestration helices as teal and other helices as yellow. Dashed lines indicate flexible and unmodelled regions. Pore entry and exit are indicated with red arrows. Lateral fenestration is depicted as a brown dashed circle with a label "Fen". **b** 2.5 Å consensus cryo-EM map of AtALMT9 from dataset 3 processed with C2 symmetry. Unsharpened map surface represents AtALMT9 and is colored in the same manner as (**a**), except brown for mixed, heterogeneous density. Unmodeled regions are colored as white. Transparent unsharpened map at a low contour level shows a micelle. Transmembrane helices helices are indicated with the corresponding numbers in circles. Blue, red

rectangles and open books indicate direction of slices and views in (**c**, **d**). **c** Cytosolic and **d** Side views of central slice of transmembrane domain. **e** A simplified illustration of (**d**) for (**f**). **f** Different datasets and observed structural classes of AtALMT9. Datasets 1 and 2 were obtained with LMNG and CHS as detergents while dataset 3 with LMNG only. Difference between the datasets 1 and 2 is that the dataset 1 was obtained in the absence of malate and the dataset 2 in the presence of malate. Observed structural classes from the dataset 1 were essentially similar to those from the dataset 2 except that no malate molecule was found in the two classes from the dataset 1. Detailed information about drawing is described on the right-most side. **g–i** Relative abundance of extracted lipids from purified AtALMT9 in dataset 3 condition using LC-MS. Each data from duplicates was represented as mean with data points shown.

"sterol 2" when two pore lipid tails of the sterol mimic (CHS) are in the fenestrations (Fig. 1f, Supplementary Figs. 3, 4). Extensive data processing of the CHS-free dataset 3 allowed us to discriminate five classes (trans1, trans2, intermediate, cis1-PI/PIP, and cis2) in the pore lipid-bound state and two classes (N and W) in the peripheral lipid-only state (Fig. 1f, Supplementary Fig. 5). We named the seven structural classes of the pore lipid-bound state based on the lipid configuration, identity, and number. In the classes "trans1" and "trans2", two tails of the pore lipid lie in two fenestrations across two protomers. In the classes "cis1-PI/PIP" and "cis2", two tails of the pore lipid are in the fenestrations of one protomer. In the class "intermediate", only one tail of the pore lipid is in the fenestration. In cis1-PI/PIP class, the pore lipid was modeled as phosphatidylinositol 4-phosphate (PI4P) solely based on electron density. We observed scant densities for polar head groups of pore lipids while a bulky inositol head group adopted a relatively fixed position in the cis1-PI/PIP class (Supplementary Fig. 7). Mass spectrometry results suggest that such weak densities stem from various head groups, but not PA with the phosphate group only (Supplementary Fig. 8). We named the two structural classes "N (narrow)" and "W (wide)" of the peripheral lipid-only state based on the pore width (Fig. 1f, Supplementary Fig. 7). The sterol mimic-bound and pore lipid-bound classes represent plugged states in which the pore is blocked by the head groups of one or two pore lipids (Fig. 1f). The peripheral lipid-only classes did not harbor pore lipids, thereby leading to unplugged states. The size of the pore is restricted by the presence of hydrophobic residues from the pore helices in the N class while it is widened by the presence of solvent and malate in the W class (Fig. 1f).

## Two distinct pore widths of peripheral lipid-only states

The high local resolution of lipids, solvent and malate densities (2.6–2.8 Å for the N class; 2.8–3.0 Å for the W class) enabled to discriminate key structural differences between the classes N and W (Supplementary Figs. 6b, 7). Both classes represent asymmetric peripheral lipid-bound states. The class N exhibits an asymmetric lipid binding in the pore (Fig. 2a). In the protomer A, we found no lipid binding in the upper fenestration ($F_{A1}$) and one short lipid tail in the lower fenestration ($F_{A2}$), whereas we found a one long tail in the upper fenestration ($F_{B1}$) and one short tail in the lower fenestration ($F_{B2}$) in the protomer B. The class W presents a more fenestrated state. We detected a long tail in the upper fenestration ($F_{A1}$) and one in the lower fenestrations ($F_{A2}$) in the protomer A, whereas we found a one short tail in the upper fenestration ($F_{B1}$) and a long lipid tail in the lower fenestration ($F_{B2}$) in the protomer B (Fig. 2b).

In addition to lipid densities, we observed solvent or malate densities in the classes N and W. In the class N, the solvent densities were located only at the pore entry and exit, reflecting the narrow pore of a region poorly accessible to the solvent (Fig. 2a, c, e). In contrast, in the class W, we observed several solvent densities inside the ion conduction pore and three malate densities at the cytosolic pore rim (S0), at the pore entry (S1), and at the pore exit (S3) (Fig. 2b, d, f, Supplementary Fig. 9). Despite the different distribution of the solvent molecules in the pore, the N and W classes exhibit common interaction features with solvents and malate. In the class N, a solvent molecule is trapped between two arginine residues, R143 and R200 of each protomer (Fig. 2c). In the class W, the R143-R200 pairs are involved in binding two malate ions: R143 and R200 from the protomer A interacted with malate at S1; R200 from the protomer A and R143 from the protomer B directly interacted with malate at S0; and R200 of the protomer B indirectly interacted with malate at S0 through a water-mediated hydrogen-bonding (Fig. 2d). W120, Y208, and R226 from each protomer interact with solvent molecules at the pore exit of the class N while two W120 and one R226 interact with malate at S3 in the class W (Supplementary Fig. 9). Y208 seems to hold R226 in the pore exit in both classes (Supplementary Fig. 9).

The peripheral lipid binding in the classes N and W seems to correlate with the side chain positioning of two basic residues - R143 and R200 - in the pore (Fig. 2). Two side chain positions of R143, referred to as "down" and "up" positions, were observed. In the class N, R143 and R200 assume the "down" position in the protomer A while they take the "up" position in the protomer B (Fig. 2a). By contrast, both R143 and R200 were in the "up" position in both protomers in the class W (Fig. 2b). Such differential positioning of the side chains of R143 and R200 was not observed in classes other than the N and W classes (Fig. 1f). The differences in the side chain positioning of the two basic residues of the protomer A between the N and W classes were substantial: ~6 Å of Cζ distance for R143; and ~3 Å of Cζ distance for R200. The interactions between R143 and R200 side chains exhibited clear differences between the N and W classes. R143 and R200 in both protomers formed a water-mediated hydrogen bond in the N class while they did not have such an interaction in the W class (Fig. 2c, d). Asymmetric side chain positioning of R143 and R200 in the N class appeared to be supported by differential occupation by the lipids of the lateral fenestrations. In the protomer B, two lateral fenestrations ($F_{B1}$ and $F_{B2}$) were occupied by lipid tails while in protomer A only one fenestration ($F_{A1}$) was occupied. A lipid tail in $F_{B1}$, along with a salt bridge between E130 and R143$^{up}$, apparently allowed the up-position of R143 (Fig. 2a, c). By contrast, two malate ions at S0 and S1 sites provide several salt bridges and hydrogen bonds, thereby supporting R143 and R200 in the up-position in the W class (Fig. 2d, f). Two salt bridges, one between E130 and R143, and the other between E196 and R200, provided additional interactions for maintaining the up-position of the two arginine residues (Fig. 2d). The two distinct side chain positions of R143 are apparently responsible for the differences in the peripheral lipid binding and the displacement of the pore helices. In the class N, the down-position of one R143 blocks the upper fenestration ($F_{A1}$) (Fig. 2a, c). The hydrophobic interactions among pore-lining residues, involving V119, W120, L123, and L204, bring together the pore helices, forming hydrophobic constriction (Fig. 2e, g). The reduced distance between two pore helices (12.2 Å and 12.5 Å of Cα distance between L123 on TM2 and L204 on TM5) and the presence of large hydrophobic residues in the hydrophobic constriction prevent peripheral lipid from entering into the pore through the lower fenestrations ($F_{A2}$ and $F_{B2}$) (Fig. 2g). A recent study suggests that AtALMT9 is gated by W120 inside the ion conduction pore[46]. Our structural data also show that W120 is located inside the pore but only shows minor displacement between the N and W classes. To understand the role of this residue we investigated the effects of alanine and phenylalanine substitutions of W120 on the transport activity (Supplementary Fig. 10). Notably, W120A/F mutations did not have major effects on the voltage-dependent activation kinetics and malate currents of AtALMT9 (Supplementary Fig. 10). Our results show that, despite its position in the pore, W120 is unlikely to act as a gate of AtALMT9 in contrast with the hypothesis proposed in the previous study[46]. In the class W, the up-position of R143 induced by malate binding opens sufficient space to allow peripheral lipids to enter in the lateral fenestrations ($F_{A1}$ and $F_{A2}$) (Fig. 2f). Subsequently, peripheral lipids have access to the hydrophobic constriction created by the separation of the two pore helices (2.2 Å and 2.2 Å increase of Cα distance between L123 on TM2 and L204 on TM5) and the displacement of the hydrophobic pore-lining residues apart from the center of the pore (Fig. 2h). The side chains of L123 and L204 swing away in a protomer, creating a hydrophilic environment instead of the hydrophobic constriction (Fig. 2h). Taken together, these observations show that the down and up positions of the side chains of R143 and R200 are key structural determinants for the width of the pore.

We observed a second major change in the positioning of the side chains of N142 and Y197 between the classes N and W. In the class N, Y197 of the protomer A protrudes into the pore while in the class W the Y197 of both protomers retracts from the pore, thereby facing TM3

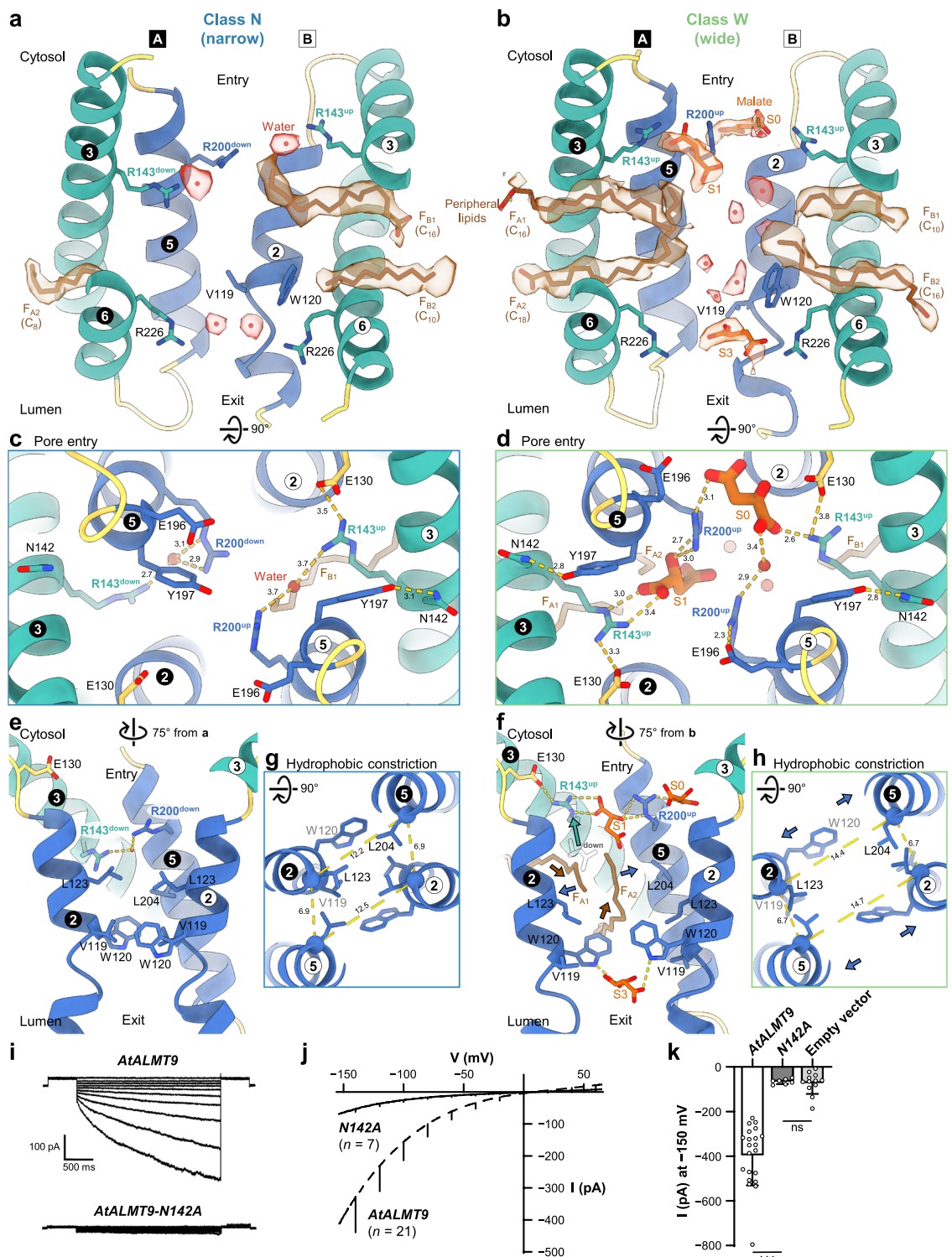

(Fig. 2c, d). The differences in the side chain positioning of Y197 of the protomer A in the classes N and W corresponds to a 9.3 Å of Oη atoms. In the class N, only Y197 of the protomer B faces TM3 and interacts with N142, whereas Y197 of the protomer A stands in the pore. In the protomer A, a slight shift of TM5 coupled with the down-position of R143 interrupts the N142-Y197 interaction (Fig. 2c). By contrast, the position of Y197 of both protomers allows a direct interaction through a hydrogen bond with N142 of TM3 in the class W. This configuration was also found in the class sterol2 where Y197 of both protomers is directed to the pore (Fig. 2c, d). We therefore hypothesized that the interaction of the N142-Y197 pair is important to lock the TMD of AtALMT9 in a wide-like conformation. Under this hypothesis, the disruption of the N142-Y197 pair would keep the TMD of AtALMT9 in a narrow-pore state. We tested this hypothesis through patch-clamp experiments on vacuoles overexpressing AtALMT9-GFP WT and mutants on N142 and Y197. We found dramatic decreases of the malate ionic currents in patches from vacuoles expressing N142A mutant when compared to WT (Fig. 2i–k). The currents in patches overexpressing N142A were the same as those recorded in the empty vector transformed vacuoles (Fig. 2i–k, Supplementary Fig. 1). In Y197A,

**Fig. 2 | Pore width determined by peripheral lipids and malate binding.**
**a–h** Transmembrane domains (TMDs) including pore and fenestration helices of the classes N and W. TMDs are depicted as cartoon with key residues in stick representations. Electron densities corresponding to water molecules, peripheral lipids, and malate molecules, locally-filtered and contoured at 6.7 σ, are shown with ball and stick representations. Only acyl chains of the peripheral lipids are modelled, presumably representing fatty acid tails with the number of carbon atoms in a chain. Pore helices are colored as blue, fenestration helices as teal, loops as yellow, lipids as brown, water molecules as red, and malate molecules as orange. The circled numbers indicate helices in protomer A (white in black) and protomer B (black in white). Distances between two atoms in Å are described along with dashed lines. Conformational states of R143 are labeled as superscripts "up" and "down".

Malate binding sites are labeled as "S0", "S1", and "S3". **a, b** Slice views of transmembrane domain around pore and fenestrations. The numbers of carbon in acyl chain or alkanes are indicated by subscript. **c, d** Cytosolic views of pore entry. **e–h** Slice views of pore and hydrophobic constriction region. **i** Representative currents from vacuolar patches overexpressing *AtALMT9* and *AtALMT9-N142A* in *Nicotiana benthamiana*. **j** Mean current-voltage characteristics from vacuolar patches overexpressing *AtALMT9* (n = 21) and *N142A* (n = 7). Error bars represent the standard deviation. **k** Mean current intensity at −150 mV in (**j**). Each data was represented as mean ± standard deviation with data points shown. Statistical analysis was done with non-parametric two-sided Mann-Whitney test. *P*-values are 0.0000017 and 0.97, respectively. *$P < 0.05$; **$P < 0.01$; ***$P < 0.0001$; ns, not statistically significant.

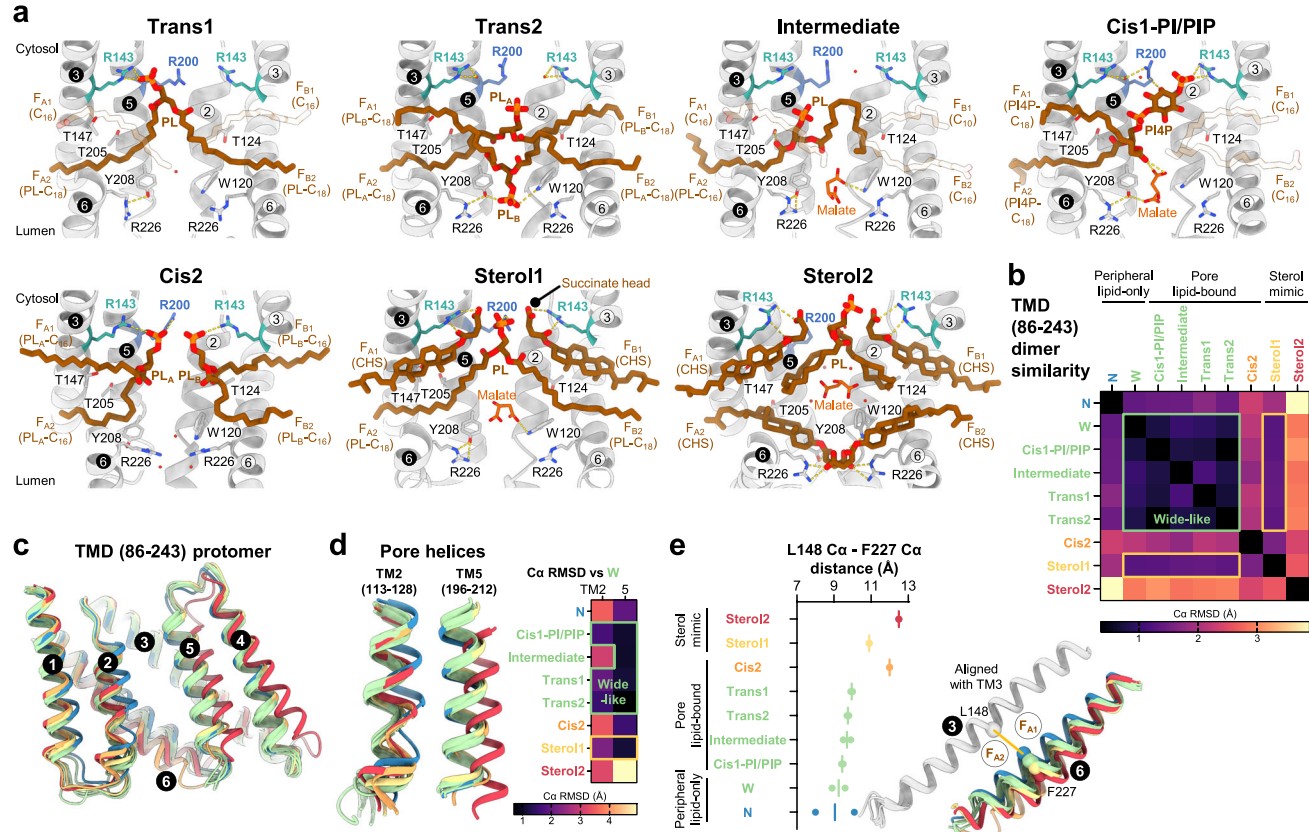

**Fig. 3 | Lipid plugs in wide pore. a** Pore and fenestrations of the pore lipid-bound and sterol mimic-bound classes. Slice views of TMDs are depicted as gray cartoon with key residues in stick representations. Pore lipids, peripheral lipids, and CHS are depicted as thick stick. Only acyl chains of the peripheral lipids are modelled, presumably representing fatty acid tails with the number of carbon atoms in a chain. Pore lipids and CHS are colored as brown, peripheral lipids as transparent brown for visibility of other lipids, water molecules as red, malate molecules as orange, selectivity filter residues R143 and R200 as cyan and blue, respectively. The numbers in circles indicate helices. The circle colors indicate protomer A (white number in black circle) and protomer B (black number in white circle). Interactions indicated as yellow dashed lines. **b** Pairwise Cα root mean square deviations (RMSDs) (Å) of TMD (86-243) dimers in all states are depicted as heatmap. Lower

RMSDs among classes are grouped and labeled as same color. Thick yellow and green border lines indicate similarity among 6 classes. **c** Superposition of TMD (86-243) protomers in all states. TMDs are depicted as cartoon representation and colored as same color in (**b**). The numbers in circles indicate helices. **d** Close-up views of pore helices in (**c**). Cα RMSDs (Å) compared to pore helices of class W are depicted as heatmap. Green and yellow boxes indicate pore helices RMSDs that are lower than indicated values (< 2.5 for TM2; < 1.5 for TM5). **e** Cα distances (Å) between L148 and F227 in all states. Points indicate L148-F227 distances of protomer. Lines indicate mean L148-F227 distances of two protomer. Two fenestration helices are depicted as cartoon representation and aligned by TM3. Fenestrations are labeled and depicted as circles. The numbers in circles indicate helices.

Y197S, and Y197F mutants we observed a dramatic decrease of malate ionic currents and the lack of malate activation in presence of chloride (Supplementary Fig. 11).

### Competitive pore binding by lipids and malate
Despite the observation of the narrow and the malate-induced wide pores in the two unplugged states, the majority of particles (~71 %) in the dataset 3 and most particles in datasets 1 and 2 represented plugged states with heterogeneous densities (Figs. 1f, 3a,

Supplementary Fig. 3-5). Unlike in class W, there were no densities corresponding to malate at the entrance of the pore in the pore lipid- and sterol mimic-bound classes (Supplementary Fig. 7). In these classes, the two residues, R143 and R200, serve as binding sites for the lipid head groups. R143 and R200 interact with the phosphate head groups of $PL_A$ in the trans1, of $PL_A$ and $PL_B$ in the cis2 classes; the 4-phosphate and inositol group of PI4P in the cis1-PI/PIP class; the succinate head groups of CHS in upper fenestrations in the sterol1 and sterol2 classes (Fig. 3a). In the intermediate and trans2 classes, we did not observe

interactions between the R143-R200 pairs and lipids (Fig. 3a). However, in these two classes the phosphate head groups cause steric clashes with malate at the S1 site (Fig. 3a). Slightly different positions of R143 and R200 also cause steric clashes with malate at S0 in all the pore lipid-bound and sterol mimic-bound classes (Fig. 3a). Despite pore lipid binding, we observed several malate densities in the S3 region in the intermediate and cis1-PI/PIP classes, and above S3 region in the sterol1 and sterol2 classes (Fig. 3a). In the remaining classes, we found several structural factors preventing malate binding: peripheral lipid of the trans1 class interferes with malate binding at S3; the cis2 class presents a narrower pore compared to other classes; phosphate head group of $PL_B$ of the trans2 class occupies S3 and interacts with W120 and Y208 at the pore exit (Fig. 3a). Similarly, the succinate head group of CHS in the lower fenestrations occupied the S3 region interacting with W120, Y208, and R226 (Fig. 3a). These observations indicate that phosphate groups of PL and the dicarboxylic moieties (malate and succinate head group of CHS) compete with each other for malate binding sites S0, S1, and S3. The high affinities for dicarboxylic group of S1 and S3 may have caused strong CHS bindings in the CHS-supplemented dataset 2.

To investigate the effects of lipid binding on the conformations of AtALMT9, we measured pairwise Cα root mean square deviations (RMSDs in Å) of TMD (residues 86-243) dimers (Fig. 3b). Despite distinct ligand binding modes, five classes share similar TMD conformations with low pairwise RMSD values (0.7–1.2 Å): the classes W, cis1-PI/PIP, intermediate, trans1, and trans2 (hereinafter called as "wide-like classes") (Fig. 3b). Interestingly, despite the presence of artificial interactions with the succinate head group in the sterol1 class, this class exhibits a small pairwise RMSD values (1.2–1.4 Å) with the wide-like classes (Fig. 3b). Unlike the sterol1 class, the other classes present a large pairwise RMSD values compared with the wide-like classes: the classes N (1.4–1.7 Å), cis2 (1.8–2.2 Å), and sterol2 (2.8–3.1 Å). We focused on the pore helices and we analyzed pairwise Cα RMSD (Å) of TM2 (residues 113-128) and TM5 (residues 196-212) (Fig. 3c, d). Consistent with TMD similarity, TM2 and TM5 of the wide-like and sterol1 classes show structural similarity with those of the class W, except TM2 of the intermediate class (Fig. 3d). We then compared the fenestration helices by measuring the Cα distances between L148 on TM3 and F227 on TM6 (Fig. 3e). The wide-like classes exhibit different peripheral and pore lipid binding, but the average distances between the fenestration helices are similar (9.3–10.0 Å) (Fig. 3e). In the class N, the distances between of L148 on TM3 and F227 on TM6 of the two protomers are different (8.0 Å and 10.1 Å) (Fig. 3e). A small difference in TMDs of the sterol1 and the wide-like classes comes from expanded space between the fenestration helices. The sterol1 class has an average distance (10.9 Å) longer than the wide-like classes due to the bulky sterol moiety, but smaller than the cis2 (12 Å) and sterol2 (12.5 Å) classes (Fig. 3e). The reduced difference between the sterol1 and the wide-like classes suggests that the sterol1 class probably resembles a native sterol binding state. Consistent with the aforementioned suggestion, the sterol1 class shows a lipid configuration similar with that of the trans1 class with sterols substituted by peripheral lipids in the two upper fenestrations. However, native sterol species are likely to have lower binding affinities compared to CHS due to the presence of a small hydroxyl head group instead of a succinate head group. In contrast, the sterol2 class is likely to represent an artificial conformation where the TMD is deformed compared to the other classes, resulting from a high affinity of the succinate group for the S3 (Fig. 3a). Taken together, these results indicate that the binding lipids induce only little conformational changes to the TMD of AtALMT9 and that upon lipid binding TMD retains the wide pore state.

## Voltage- and malate-dependent conduction through unplugged pores

Our cryo-EM structures, obtained at zero membrane potential, presumably represent a non-conductive, basal conformational ensemble at near-zero tonoplast potential such as those found in closed stomata[9]. We analyzed the pore radii using MOLE server[47] to identify differences in the basal state pores depending on lipid binding states (Fig. 4a–c). The N class reveals a long narrow pore with two constrictions: one between L123 and L204 (0.9 Å radius); and the other between V119 and W120 (1.2 Å radius) (Fig. 4a). The two constrictions are too narrow to allow the passage of a water molecule or a chloride ion (1.8 Å radius) (Fig. 4c). In the W class, the structural changes induced by malate binding and the presence of peripheral lipids result in a loosening of the two constrictions through a separation of the pore helices (Fig. 4b). The constriction from L123 and L204 is widened up to a 3.9 Å radius, which would be enough for a hydrated chloride ion or a malate to pass through. The second constriction between V119 and W120 is widened to 2.5 Å (Fig. 4c). The observation of solvent densities in the pore corroborates an increased solvent accessibility of the widened pore in the W class (Fig. 4b). Despite the negligible structural differences among the wide-like classes, all the plugged classes appear to show blocked pores (Fig. 4d, e). In the pore lipid-bound classes, either S1 or S3 is occupied by different moieties of the pore lipids. In the trans1 and cis2 classes, a phosphate head group occupies S1; in the trans2 class, S3 is occupied by a phosphate head group of $PL_B$; in the cis1-PI/PIP class, the 4-phosphate group of PI4P is in S1; and a lipid tail fills S1 in the intermediate class (Fig. 4d, e). Without the succinate head group of CHS, a phosphate head group is expected to occupy S1 in the sterol1 class, like in the trans1 class (Fig. 4d). Hydrophobic constrictions are also occupied by other moieties of the pore lipids such as a tail or an acyl group of PL (Fig. 4d, e). These observations indicate that all the pore lipids act as plugs blocking the pore.

To probe conductive states in AtALMT9, we performed molecular dynamics (MD) simulations using the unplugged narrow and wide structures (Fig. 2a, b, Fig. 4f–h, Supplementary Fig. 12a-c, Supplementary Table 2, Supplementary Movie 1-9). The TMD (residues 77-248) region in the narrow or wide class was inserted into the 1-palmitoyl-2-oleoyl-sn-glycero-3-phosphocholine (POPC) bilayer membrane after peripheral lipids were replaced by POPC. Then, we put the protein-lipid system in a solution with different anionic conditions: 200 mM chloride ($[Cl^-]^{High}$), 200 mM malate ($[malate]^{High}$), or 200 mM chloride plus 3 malate molecules ($[Cl^-]^{High} + [malate]^{Low}$) in the W class. For each setup, we performed MD simulations for about 3 microseconds under a cytosolic negative membrane potential of 0 or −500 mV.

In the simulations at zero membrane potential, no anion flux was observed both in the N and W classes (Supplementary Fig. 12a–c, Supplementary Movies 1–3). Chloride or malate was found preferably at S1 and S3 in the unplugged classes, as in the cryo-EM structures (Fig. 4a, b, Supplementary Fig. 12a, b). The N class with chloride ion ($[Cl^-]^{High}$) showed negligible fluctuations in positions of the pore-lining arginine residues (R143, R200, and R226) at zero membrane potential (Supplementary Fig. 12a, d). By contrast, the W class with chloride ion and three malate molecules ($[Cl^-]^{High} + [malate]^{Low}$) exhibited dissociation of shallowly-inserted peripheral lipid tails from fenestrations. Consequently, R143 residues showed downward movements due to the absence of tails beneath it (Supplementary Fig. 12b, e).

In the simulations at a cytosolic negative membrane potential, we observed that conformations of both N and W classes in the basal state converged into those in the conductive states representing both $R143^{down}$ and $R226^{up}$ (Fig. 4f–h, Supplementary Fig. 12d–f, Supplementary Movies 4–9). Such opposing movements of R143 and R226 rotamers allowed a new anion binding site (S2) between S1 and S3 by relieving hydrophobic constrictions of the pore in the basal state (Fig. 4f–h). At high malate concentrations ($[malate]^{High}$), three malate molecules occupied S1, S2, and S3. An entering malate pushed a pre-existing malate into the next site and a malate at S3 exited (Fig. 4f). Such ion transport is reminiscent of a knock-on mechanism in potassium channels[48]. Such knock-on effects induced a continuous flow of

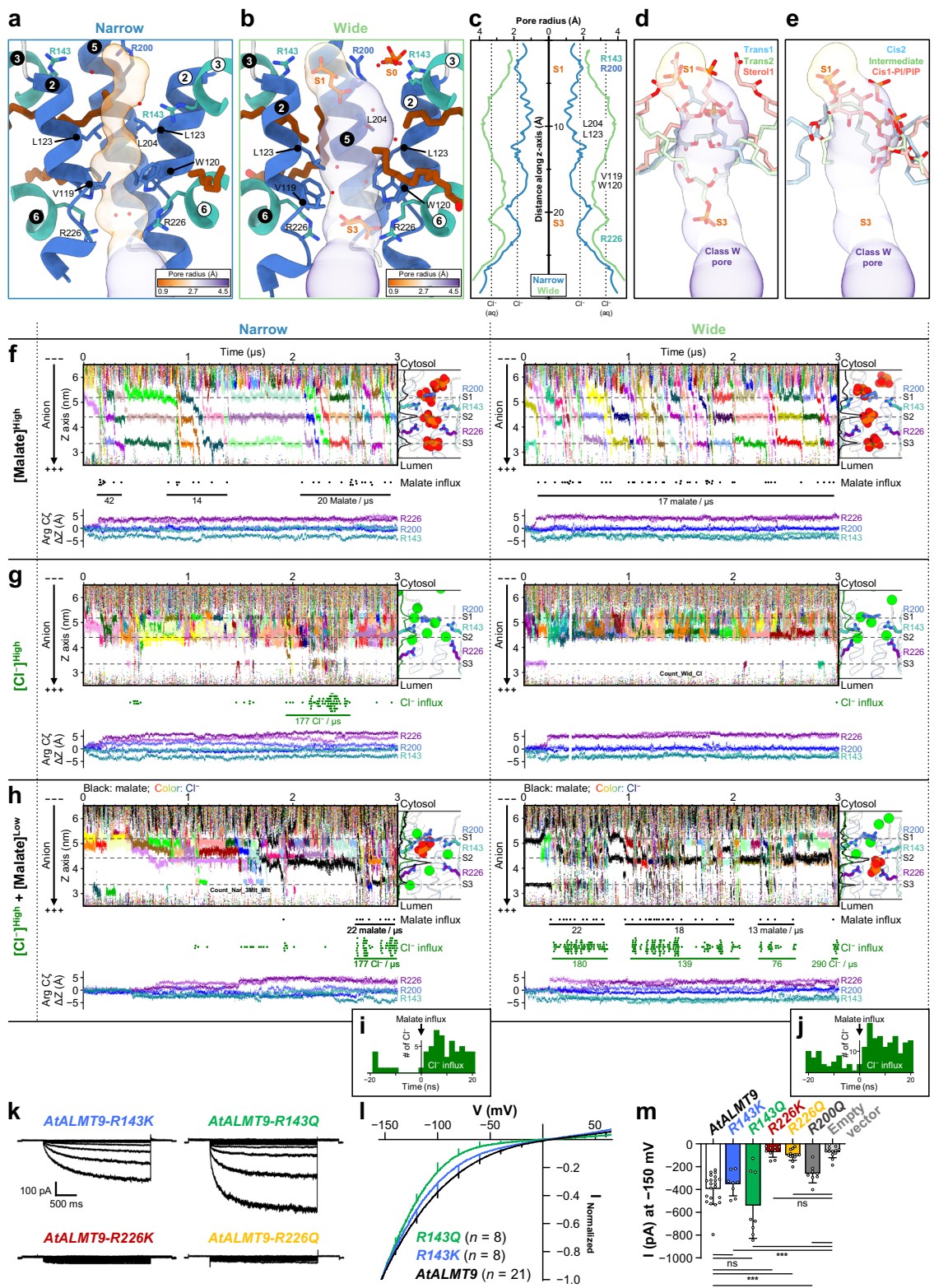

malate ions. By contrast, chloride ions represented a sporadic influx due to chloride ion trapping among R143, R200, and R226 at S1 and S2 at high chloride concentrations ([Cl⁻]^High) (Fig. 4g). In presence of a small portion of malate ([Cl⁻]^High + [malate]^Low), a malate entering S2 disrupted chloride retention in S1 and S2, thereby allowing the subsequent passage of many chloride ions (Fig. 4h–j). When comparing simulations ([Cl⁻]^High vs. [Cl⁻]^High + [malate]^Low), the low malate concentration only increased the frequency of the conductive phase

(Fig. 4g, h). In each simulation, both average transit times (20–26 ns) and average flow rates (142–177 ions per µs) of chloride ions were similar during conductive phase, regardless of malate ions (Fig. 4g, h). These simulation results are consistent with the malate-dependent open probability increase without conductivity changes in single-channel analysis[9].

Our present findings indicate that R143, R200, and R226 are important for anion permeation. It has been reported that R143E/N,

**Fig. 4 | Voltage- and malate-dependent anion conductance of narrow and wide pores. a, b** Conductance pathways of the classes N and W. Cartoon and coloring scheme is the same as in Fig. 2a, b. The circled numbers indicate helices. Malate binding sites are labeled as "S0", "S1", and "S3". Transparent pore surface are colored by pore radii. **c** Pore radius plot of the classes N and W. Pore-lining residues are labeled on corresponding positions. Radii for ionic (Cl⁻) and hydrated chloride ions (Cl⁻ (aq)) are indicated by vertical dot lines. **d, e** Pore lipids of several classes. Lipids are depicted as stick representation. Surfaces of the class W pore are used to visualize the pore. **f–h** Anion trajectory plots along the pore (z-axis) of AtALMT9 TMD (77-248) in MD simulations at −500 mV. Each panel is identified with the unplugged class and anion composition. Representative anion distributions and arginine configuration in the pore are depicted on the right side of each panel: cartoons for protein; line graphs for anion distributions; sticks for key arginine residues; green spheres for chloride; red and orange spheres for malate; gray dashed lines for "S1", "S2", and "S3". A dot plot below the trajectory indicates the

individual anion influx. Average anion flow rates are labeled with the horizontal lines. The bottom side graphs indicate positional changes of key arginine residues (Cζ) relative to initial positions. **i, j** The relative time distribution plots of chloride ion influx according to malate influx during simulations in (**h**). **k** Representative current profiles from vacuolar patches overexpressing *AtALMT9-R143K*, *R143Q*, *R226K*, and *R226Q*. **l**, Normalized current voltage characteristics from vacuolar patches overexpressing *AtALMT9*, *R143K*, and *R143Q*. Error bars represent the standard deviation. **m** Mean current intensity at −150 mV. *AtALMT9* ($n = 21$), *R143K* ($n = 8$), *R143Q* ($n = 8$), *R226K* ($n = 9$), *R226Q* ($n = 13$), and *R200Q* ($n = 8$). Each data was represented as mean ± standard deviation with data points shown. Statistical analysis was done with non-parametric two-sided Mann-Whitney test without correction for multiple comparisons. *P* values are 0.58, 0.32, < 0.000001, < 0.000001, 0.0059, 0.000016, 0.000064, 0.70, 0.087, and 0.000064, respectively. *$P < 0.05$; **$P < 0.01$; ***$P < 0.0001$; ns, not statistically significant.

R200E, and R226E/N completely abolish ion currents compared to WT, except R200K/N[49]. Such observations suggest the importance of the charge and size of these residues in ion transport by AtALMT9. Subsequently, we performed vacuolar patch clamp recordings expressing R143K/Q, R200Q, and R226K/Q mutants (Fig. 4k–m, Supplementary Fig. 13, 14). The glutamine substitution removes the positive charge but can interact with malate, and features a side chain longer than asparagine. R143K mutant, keeping the positive charge but possessing a different shape, behaved similarly to WT (Fig. 4k–m, Supplementary Fig. 13a–d). R143Q mutant, which has no positive charge and a longer side-chain compared to asparagine, showed malate currents similar to those of WT but a faster activation kinetics and a voltage dependency shifted to more negative values (Fig. 4k–m, Supplementary Fig. 13a-d). R200Q mutant also showed a similar tendency similar to WT (Supplementary Fig. 13f–i). The charge neutralization in these mutants presumably decreases the interaction with anions, rendering the anions requiring more energy for transit. Consistent with this interpretation, R200N exhibits more negative membrane potential activation[49]. R200 is located on the pore helices, so its side chain length is expected to have less impact on the interaction with the anions than R143. Notably, unlike R143 and R200, R226K/Q mutants were unable to conduct malate currents (Fig. 4k, m, Supplementary Fig. 13e). In the simulations, the application of a cytosolic negative membrane potential causes a R226 movement toward the cytosolic side (Supplementary Fig. 12d–f). Since all tested R226 substitutions disrupt ion conduction, both charge and shape of R226 seem to be crucial for AtALMT9 functionality. Taken together, the results from MD simulations and vacuolar patch clamp recordings corroborate the importance of R143, R200 and R226 in ALMT9 activation.

## Conduction pore plugging by lipids through lateral hydrophilic fenestration

The majority of ALMT9 conformations at the basal state are plugged with lipids (Fig. 1), indicating that channel activation requires a transition from the plugged states to unplugged states. The movement of pore lipids in and out of the pore should occur during channel activation. Such movements of hydrophobic lipid tails in and out of TMD through the fenestration are possible since it is positioned in the middle of lipid bilayer (Figs. 1a, b, 5a). However, the hydrophilic phospholipid head group movements across the hydrophobic membrane and fenestration are thermodynamically unfavorable and unlikely to occur without structural assistance such as a hydrophilic cleft as reported in scramblase[50,51]. Therefore, we analyzed the lipophilicity of the TMD surface of the N, W, and cis2 classes to unveil how pore lipids access the middle of the membrane. We found several hydrophilic surfaces throughout fenestration and its periphery (Fig. 5a–c, Supplementary Fig. 15). Firstly, the two fenestration helices form hydrophilic surfaces that penetrate the membrane (Fig. 5a, Supplementary Fig. 15). The hydrophilicity of membrane-facing side mainly

comes from one serine and eight glycine residues: five glycine residues (G146, 151, 152, and 156) and S149 on TM3, G198 on TM5 and two glycine residues (G235 and 239) on TM6 (Fig. 5a). Owing to the absence of a side chain, the main-chain carbonyl oxygen and amide nitrogen atoms of glycine residues are exposed, forming hydrophilic surfaces. Secondly, hydrophilic patches are located inside the fenestration (Fig. 5b, c). The upper side patch (facing TM3 and 5) is bigger and more hydrophilic than the two patches on the lower side (facing TM2 and 6) (Fig. 5b, c). The hydrophilic patches are formed by the main chain atoms of G140 on TM3 and of three threonine residues (T124 on TM2, T147 on TM3, and T205 on TM5) (Fig. 5b, c). Hydrophilic acyl group of the intermediate and cis1-PI/PIP classes interacts with or is juxtaposed to T205 of the upper side patch (distance between Oγ1 of T205 and oxygen atom of acyl group: 3.1 Å in the intermediate class and 3.8 Å in the cis1-PI/PIP class, respectively) (Fig. 3a). Lastly, the selectivity filter residues R143 and R200 serve as hydrophilic binding sites for lipid head groups (Figs. 3a, 5b, c). These results collectively demonstrate the presence of three separate hydrophilic surfaces in the membrane side, fenestration, and pore entry of ALMT9.

To investigate whether the hydrophilic surfaces of ALMT9 are conserved among ALMT family proteins, we superposed the N and W classes structures of AtALMT9 with known open state structures of GmALMT12 (PDB: 7W6K) and AtALMT1 (PDB: 7VQ7, pH 5, Al³⁺) (Supplementary Fig. 16). We also made multiple sequence alignments of ALMTs with known voltage-dependent kinetics: AtALMT9[9], AtALMT4[23], AtALMT6[7], and VvALMT9[11] for slow activation of clade 2; GmALMT12[8] and AtALMT12[52] for rapid activation of clade 3; and AtALMT1[21] and TaALMT1[53] for instant activation of clade 1 (Fig. 5d–f, Supplementary Fig. 17). The membrane-facing side and pore entry differs in three clade ALMTs (Fig. 5a–f, Supplementary Fig. 16f-i, 18a, b). R200, important for lipid binding, is conserved in clade 2, but the corresponding residues of clades 1 and 3 have short hydrophilic and aliphatic side chains (Fig. 5d). Clade 2 ALMTs have more glycine, serine, or threonine residues on the membrane-facing side of TM3 and TM6 than clade 1 and 3 ALMTs (Fig. 5e, f). Instead of glycine residues in the clade 2, bulky aromatic or large aliphatic residues occupy the membrane side in the clades 1 and 3 (Fig. 5e, f). These differences apparently contribute to making hydrophilic surfaces on the membrane side of TM3 and TM6 in the clade 2, and hydrophobic surfaces in the clades 1 and 3 (Supplementary Figs. 16f–i, 18a, b). Despite differences in the membrane side and pore hydrophilicity, the fenestration interior exhibits conserved patterns of glycine and threonine residues in all clades (Fig. 5e, f, Supplementary Fig. 16a–d, f–i). In GmALMT12 of the clade 3, each protomer has two unidentified densities in large fenestration (hereinafter called as $F_{A1}$ and $F_{A2}$)[8] (Supplementary Figs. 16c, 18a, c, d). These densities are likely to have derived from lipids (Supplementary Fig. 18e). The conserved glycine and threonine residues are likely to be involved in fenestration. Other conserved or unique glycine residues in the fenestration helices are related to close contacts toward

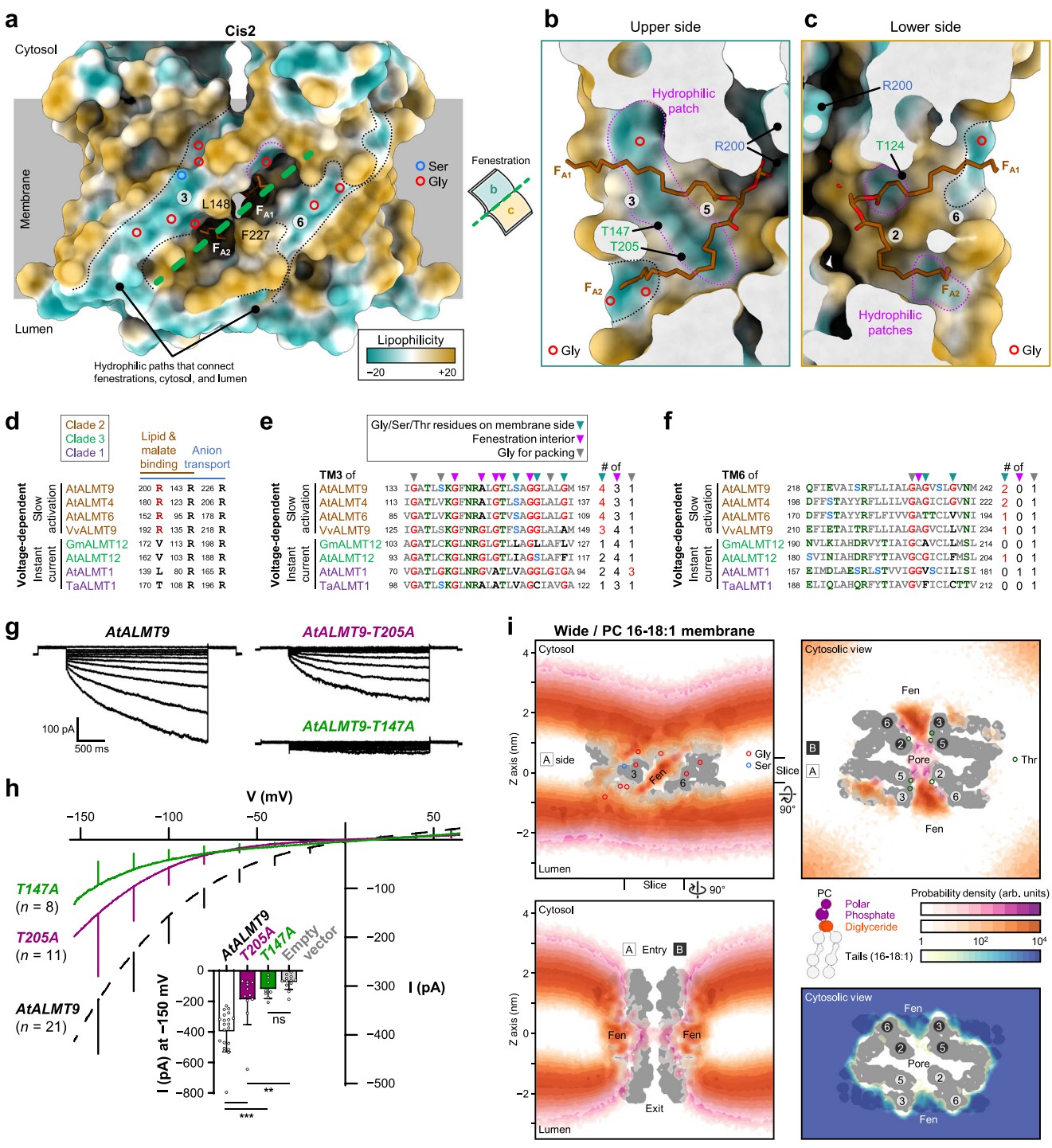

pore helices or Trp-Glu-Pro (WEP) motif of the intracellular domain that is crucial for function in GmALMT12[8] (Fig. 5e, f, Supplementary Fig. 16e–h). AtALMT1 of the clade 1 possesses no fenestration, and the conserved threonine residues apparently stabilize the selectivity filter. T61 directly interacts with R80; T84, T144, and W57 (W120 of AtALMT9) make a hydrophilic pocket for positioning of R165 (Supplementary Figs. 16d, 18b).

To validate the functional roles of the threonine residues constituting the hydrophilic patches in fenestration, we prepared T147A and T205A mutants and analyzed its functionality by electrophysiology (Fig. 5g, h). We found that patches expressing T147A displayed abolished malate currents and that T205A showed significantly reduced malate currents (Fig. 5g, h). T147A and T205A seem to disrupt

the hydrophilic connection between the lipid head group binding site on the pore and the hydrophilic paths located outside of the fenestration (Fig. 5b). These data support the role of T147 and T205 that play an essential function in AtALMT9 and, together with the structural data, highlight the importance of the hydrophilic fenestration in the functioning of AtALMT9.

To corroborate the lipid migration through hydrophilic fenestration at an extended time scale, we conducted coarse-grained simulations using the Martini3 force field[54] for dozens of microseconds with the TMD structures in the narrow, wide, and cis2 classes (Fig. 5i, Supplementary Fig. 19, Supplementary Table 3, Supplementary Movie 10-12). We observed that hydrophilic moieties of lipids intruded the hydrophobic layer of the membrane along hydrophilic TM3, but

**Fig. 5 | Pore plugging through hydrophilic lateral fenestration. a–c** Lipophilicity around the fenestration regions. Surfaces of the Cis2 class are colored by lipophilicity. The numbers in circles indicate helices. Hydrophilic surfaces are indicated by dot lines. Fenestration and residues are labeled. Glycine and serine residues are indicated by colored circles. Green dashed line and open book indicate direction of clipped views (**b**) and (**c**). **a** Side view of transmembrane domain. **b, c** Clipped views of fenestration. **d–f** Key residues from multiple sequence alignment results of various ALMTs with known voltage-dependent kinetics. The name of ALMTs are colored by clades. Key residues are indicated by inverted triangles (teal for membrane side; magenta for fenestration interior; grey for glycine for packing). Each residue of TM3 and TM6 are colored as red for glycine; blue for serine; green for other hydrophilic residues. **g** Representative currents from vacuolar patches overexpressing *AtALMT9*, *T147A*, and *T205A* in *Nicotiana benthamiana*. **h** Mean current-voltage characteristics from vacuolar patches overexpressing *AtALMT9*, *T147A*, and *T205A*. Error bars represent the standard deviation. Inset, Mean current intensity at −150 mV. Each data was represented as mean ± standard deviation with data points

shown. Statistical analysis was done with non-parametric two-sided Mann-Whitney test without correction for multiple comparisons. *P*-values are 0.00010, < 0.000001, 0.0036, and 0.13, respectively. *$P < 0.05$; **$P < 0.01$; ***$P < 0.0001$; ns, not statistically significant. **i** Probability distributions for the lipid moieties in the coarse-grained simulation. Top left, clipped side view as shown in (**a**). Bottom left and top right, clipped side and cytosolic views rotated 90 degrees along the indicated directions for the hydrophilic polar, phosphate, and diglyceride moieties. Lower right, clipped cytosolic view for the hydrophobic lipid tails. Probability of each lipid moiety is depicted using the indicated color gradient (center right). The membrane lipid used in the simulation is phosphatidylcholine (PC) with symmetric 16-18:1 tails. Only the wide (W) class are shown. For other classes, see Supplementary Fig. 19. The boxed or circled numbers indicate protomers or helices in protomer A (white in black) and protomer B (black in white). The positions of fenestrations (Fen) are labeled. Glycine, serine, and threonine residues are indicated by circles with indicated colors.

not near the TM6 (Fig. 5i). Both the hydrophilic moieties and hydrophobic tails entered the fenestration in coarse-grained simulations, regardless of the TMD conformation (Fig. 5i). Unlike restricted pore access of lipids in unplugged states, the plugged cis2 class showed high occupancy of the pore lipids in the pore regions, as observed in the cryo-EM structure (Supplementary Fig. 19a, b). We also performed simulations with different membrane compositions according to tail lengths and saturation degrees (Supplementary Fig. 19c, d). The membrane thickness and stiffness seem to affect lipid migration toward the fenestration. The hydrophobic layer disruption of the membrane was unlikely to occur in the lipid bilayer of long saturated lipids owing to the dense and thick hydrophobic layer (Supplementary Fig. 19c, d). Contrary to the cryo-EM structures, we observed high occupancy of the hydrophilic moieties in the fenestration in simulations. It may come from different hydrophobic layer properties of lipid bilayer and micelle. Taken together, these results suggest that the hydrophilic lateral fenestration is likely to mediate the lipid migration for channel modulation.

## Discussion

Our data provides important molecular details on the pore residues of AtALMT9 together with more functional data than any other members of the clade 2 ALMTs[9,11,49,55]. AtALMT9 is permeable to both monovalent inorganic chloride ions and divalent organic malate ions[9]. In guard cells, AtALMT9 mainly functions as a chloride channel that is activated by cytosolic malate[9]. AtALMT9 is blocked by trivalent citrate ions[49] and ATP[56]. A previous mutagenesis analysis targeting conserved residues of the TMD identified positively charged residues (lysine and arginine) putatively forming the pore region of AtALMT9[49]. Our cryo-EM structures of AtALMT9 uncover conformational changes of key residues – three pore residues (R143, R200, and R226) and two cytosolic lock residues (N142 and Y197) in activation (Fig. 2). The three arginine residues face the pore and form two malate binding sites, one at the cytosolic entrance (R143 and R200) and the other at the vacuolar exit (R226) of the permeation pathway. The pair N142-Y197 interacts through hydrogen bonding when R143 is in the upward position, while when R143 is downward N142 and Y197 are separated and do not interact (Fig. 2c). Patch-clamp analysis of AtALMT9 mutants targeting N142 and Y197 to abolish the interaction, confirmed its importance for maintaining AtALMT9 in a wide-like conformation (Fig. 2i). These observations suggest that the R143, R200, and R226 together with the N142-Y197 pair constitute key conformational determinants for AtALMT9 activation.

Our MD simulations data shed light on the voltage-dependent functionality of AtALMT9. The cryo-EM structures of AtALMT9 most likely represent basal states at zero transmembrane potential. MD simulations applying a transmembrane potential revealed that the movements of side chains of R143 and R226 along with the removal of

peripheral lipids contribute to the ion permeation (Fig. 4). Our MD simulation results are in line with experimentally observed properties of AtALMT9. Since 1 pA is equivalent to the transit of $6.24 \times 10^6$ monovalent ions per sec[57], we calculated that the chloride ion flow rates (142–177 ions per μs) in our simulations correspond to −22.8 to −28.4 pA at −500 mV (Fig. 4f–h). Single channel currents deduced from the MD simulations are in the range of −3.64 to −4.54 pA at −80 mV, consistent with the experimentally measured single channel currents of −2.7 to −2.8 pA at −80 mV[9]. Movement of R143 and R226 side chains in the ion permeation in conjunction with AtALMT9 functionality is corroborated by patch-clamp analysis using R143 and R226 mutants. While R143E, R143N, and all R226 mutants disrupt AtALMT9 functionality[49], we found that R143K and R143Q do not disrupt the channel functionality (Fig. 4k–m). It is noteworthy that R143Q modifies the voltage dependency of AtALMT9 and activation kinetics (Fig. 4l, Supplementary Fig. 13a–d). Taken together, the data obtained by the MD simulations show ion transport through the pore and provide significant insight into the voltage-dependent changes occurring in AtALMT9.

The presence of lipids and sterols in the pore of our structure suggests that they might regulate the activity of AtALMT9 in vivo. We observed densities corresponding to lipids in all nine classes of AtALMT9 structure (Supplementary Fig. 7). In mass spectrometry, we detected the existence of zwitterionic, anionic lipids, and PIs (Supplementary Fig. 8). These lipids are also present in the vacuolar membrane[58–60] and the PIs have been reported to regulate stomata closure[60]. The negatively charged moieties of the pore lipids are trapped by the positively charged R143-R200 pair (Fig. 3). In the pore lipid-bound classes, we observed densities that could be fitted to various head groups. This diversity in head group densities implicates that the head groups affect the binding pose of phospholipids. The diverse binding configurations and physicochemical properties according to the head group may then have different effects on the channel activity. Once in the pore, the lipids act as a plug blocking the permeation pathway while they induce negligible structural changes between plugged and unplugged classes (Fig. 3). Blocking of the pore by anionic lipids in the pore lipid-bound classes can be a key for the regulation mechanism of the clade 2 ALMTs. It seems that pore lipids need to be kicked out of the pore to make it accessible to chloride ion or malate. We found the hydrophilic surfaces in the fenestration interior and at the periphery that may allow pore lipid movement (Fig. 5). The hydrophilicity of these regions comes from exposed main chain atoms of glycine residues or the hydroxyl groups of serine and threonine residues (Fig. 5). Interestingly, sequence alignment suggest that this stepping hydrophilic pathway is conserved only in the clade 2 ALMTs (Fig. 5). These results could suggest that plugging/unplugging the pore from lipids may be involved in the regulation of ALMT9 ion transport activity and possibly in other ALMTs of the clade 2.

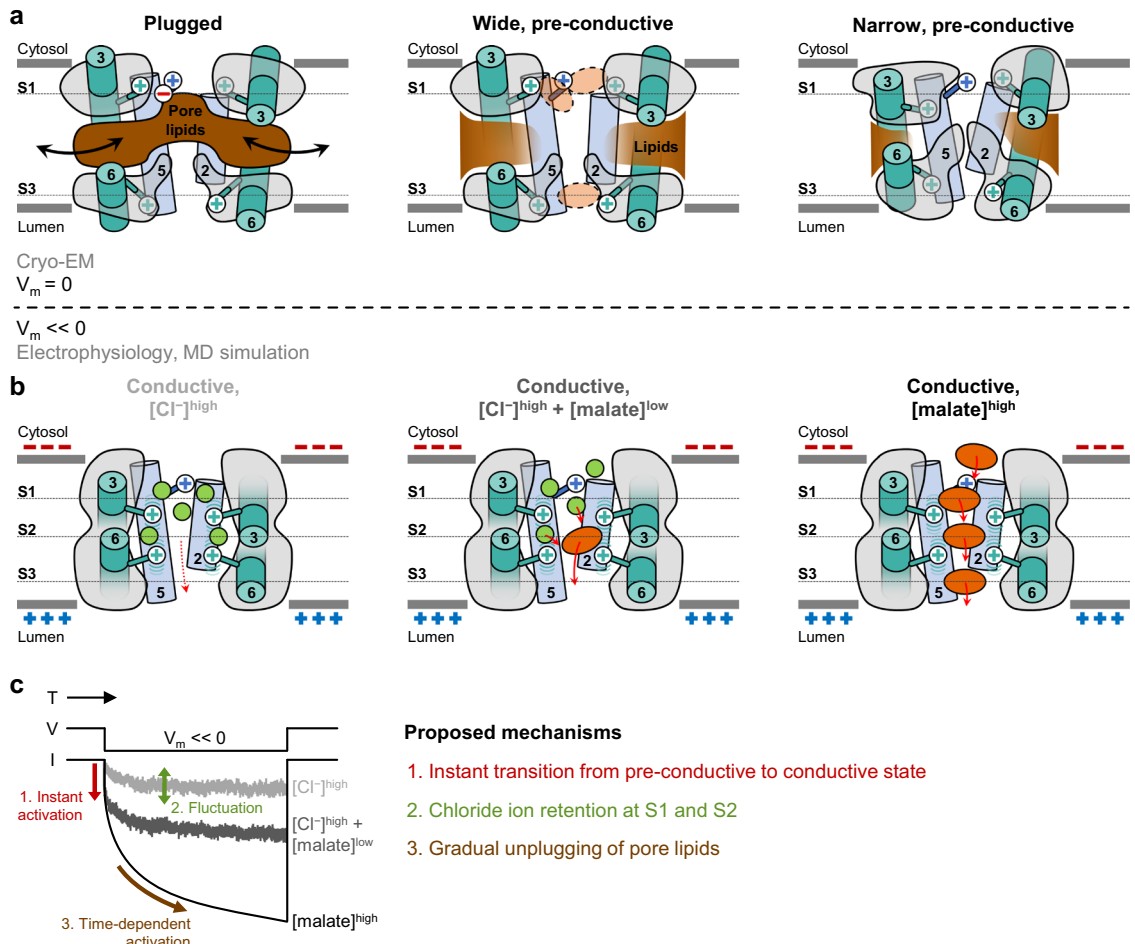

**Fig. 6 | A proposed malate-driven lipid unplugging mechanism of AtALMT9. a** Putative basal conformational ensemble observed in cryo-EM. **b** Putative voltage-dependent anion conductive states based on electrophysiology and MD simulations **c** Proposed mechanisms in voltage-dependent current according to different anion compositions.

Additionally, the TMD of the sterol mimic-bound classes is similar with those of the pore lipid-bound and peripheral lipid-bound classes, suggesting that plant sterols may contribute to the regulation of the clade 2 ALMTs.

We envisage a prospective mechanism of AtALMT9 based on our structural, simulation and electrophysiological data. At nearly zero tonoplast potential, AtALMT9 populates three basal pore states with different conductivity: a non-conductive state plugged by various pore lipids and two pre-conductive unplugged states with wide and narrow pores (Fig. 6a). Hypothetical voltage-dependent conductive states are permeable to chloride and malate ions (Fig. 6b). The pre-conductive wide and narrow states in small populations contribute to instantaneous activation upon application of cytosolic negative membrane potential, as in stomata opening. In the conductive states, chloride ions are retained at S1 and S2 by arginine residues, which induces fluctuation of the ion fluxes through the pore (Fig. 6b, [Cl⁻]$^{High}$). Low concentrations of malate enhance ion flux by disrupting chloride ion trapping (Fig. 6b, [Cl⁻]$^{High}$ + [malate]$^{Low}$). High concentrations of malate induce a "knock-on" effect, reducing the fluctuation induced by the anion trapping at S2 site (Fig. 6b, [malate]$^{High}$). The transitions to the conductive state need a much shorter time (less than microseconds) than observed voltage-dependent kinetics (Fig. 4f–h, Supplementary Fig. 1). The prevalence of the plugged states and gradual unplugging of pore lipids are plausible to explain these slow activation kinetics (Figs. 5i, 6c).

Recently, three AtALMT9 structures have been deposited in PDB: apo state at a resolution of 3.3 Å (PDB: 8HIW, EMD-34828) and two malate-supplemented states at 3.8 and 3.6 Å resolutions (pH 7.5, PDB: 8HIY, EMD-34829; pH 5.0, PDB: 8ZVF, EMD-60510)[46]. The cryo-EM maps of apo- and malate at pH 7.5 structures have heterogeneous, mixed lipid traces that were observed in our consensus map (Supplementary Fig. 20a, b). The structure with 10 mM malate at pH 5.0 has a phosphatidic acid-shaped density in the lateral fenestration[46]. However, heterogeneous lipid-bound states are difficult to discriminate with relatively lower resolution maps. For these reasons, these reported structures have very similar TMD conformations with our wide and wide-like structures (less than 1.5 Å Cα RMSD), but distinct from TMD of the class N (1.8 and 1.7 Å Cα RMSD with apo and malate-supplemented states, respectively). Pore-lining residues and pore radii represent similar configuration with wide pore of the class W (Supplementary Fig. 20c–e). Nevertheless, these results also support the majority of wide-like states and the presence of lipid densities.

Despite of extensive results, our study has certain limitations. The membrane of the heterologous expression system[45] differs from plant vacuoles[58–60], thereby raising a possibility that some plant-specific lipids may have been ignored. Lipid preference or specificity of each plugged class in indeterminate due to the ambiguity of density, except the cis1-PI/PIP class. Addition of a detergent LMNG may have introduced artifacts in protein purification. Notably, the purification of GmALMT12/QUAC1 whose structure has fenestration employed LMNG for purification[8]. To check whether lateral fenestrations are artifacts from our purification protocol, we determined the cryo-EM structure

of AtALMT1 with the same purification method as with our AtALMT9 (Supplementary Fig. 21a–e). We observed no significant differences between the previously reported AtALMT1 structure with GDN (PDB: 7VQ4)[21] and our AtALMT1 structure with LMNG and CHS (Supplementary Fig. 21f). Reported AtALMT9 structures with digitonin or GDN (PDB: 8HIW, 8HIY, 8ZVF)[46] also showed negligible differences with our AtALMT9 structures with LMNG (Supplementary Fig. 20). These results suggest that our detergent system has little impact on observed structures, especially the fenestration region. Nonetheless, some questions remain regarding the physiological and functional relevance of lipids in channel regulation.

The structural and functional results we obtained on AtALMT9 provide understanding about the broad functional spectrum of the ALMT family. Structural analysis and multiple sequence alignments identified key residues in the ion conduction pathways that are conserved in the three ALMT structures but were observed in different conformations. Mutations of these key residues cause severe impairments of the ion transport capacities. The conservation of these residues through all ALMTs shows that their functional importance avoids their mutation during the evolution of this protein family. Interestingly, the conformational variability of the fenestration helices and of wide pore region may allow a structural diversification to fulfill functional niche. On the other hand, the results we obtained raise challenges regarding the identification of diverse structural repertoires and functions within the ALMT family. Each ALMT has evolved unique regulatory mechanisms and conductive properties within clade 2[7,9,11,23]. However, different regulatory mechanisms resulting from different conformation of distinct sites are difficult to predict with limited structural information[8,21]. The interaction of fenestration helices with lipids is almost impossible to predict by the AlphaFold-based structural predictions[8,22]. Therefore, further structural studies are indispensable for investigating the functional characteristics of the ALMT family.

## Methods

### Constructs and recombinant baculovirus production
Full-length AtALMT9 (At3g18440, UniProt entry: Q9LS46) or AtALMT1 (At1g08430, UniProt entry: Q9SJE9) gene from *Arabidopsis thaliana* complementary DNA (cDNA) was amplified by PCR. To produce a recombinant baculoviral genome, AtALMT9 gene was cloned into modified pACEBac2-sfGFP-HA-H$_{10}$ (Geneva Biotech) vector[61]. Baculoviruses were generated as previously described[61]. Briefly, the recombinant baculoviral genome was generated by transformation of vectors into chemically competent *E. coli* DH10EmBacY cells (Geneva Biotech). Blue-white screening and colony PCR were used to select a bacmid harboring the AtALMT9 gene cassette. Baculoviruses were generated by bacmid transduction into Sf9 cells (Gibco) (at low passage, $2.5 \times 10^6$ cells mL$^{-1}$, 30 mL Sf-900 II SFM (Gibco), > 90 % viability) using ExpiFectamine™ Sf transfection reagent (Gibco). For higher titer, P0 viruses were used to produce P1 viruses. Baculoviruses were supplemented with 0.1 % (w/v) bovine serum albumin (BSA) and filtrated by 0.2 μm cellulose acetate filter.

### Protein expression and purification
Protein purification was performed as previously described in ref. 61 with a few tweaks. Recombinant proteins were expressed in Sf9 cells ($2 \times 10^6$ cells mL$^{-1}$ in 500 mL Sf-900 II SFM) at 27 °C for 3 days using P1 viruses at a multiplicity of infection of 10.0. Sf9 cells containing recombinant AtALMT9 or AtALMT1 were harvested and stored at −80 °C. Harvested cells or pellets were resuspended sequentially in hypotonic buffer [20 mM HEPES pH 7.6, 1 mM Tris (2-carboxyethyl) phosphine (TCEP), 0.25x Pierce™ EDTA-free protease inhibitor (Thermo Fisher Scientific)], hypertonic buffer [50 mM HEPES pH 7.6, 1 M NaCl, 1 mM TCEP, 5 % (v/v) glycerol, and 0.25x Pierce™ EDTA-free protease inhibitor (Thermo Fisher Scientific)] and solubilization buffer

[50 mM HEPES pH 7.6, 150 mM NaCl, 20 mM imidazole pH 8.0, 1 mM TCEP, 5 % (v/v) glycerol, 0.5x Pierce™ EDTA-free protease inhibitor (Thermo Fisher Scientific), 2 % (w/v) *n*-dodecyl β-ᴅ-maltoside (DDM), with or without 0.2 % (w/v) cholesteryl hemisuccinate (CHS), 2 mM MgCl$_2$, and 12.5 U mL$^{-1}$ benzonase]. Each resuspension step was conducted by gentle dounce homogenization followed by centrifugation. The solubilized lysate was stirred at 4 °C for 3 h and centrifuged at 22,000 $g$, 4 °C for 1 h. The recombinant proteins were captured by stirring at 4 °C for overnight in 10 mL pre-equilibrated TALON Superflow resin (GE Healthcare). Econo-Column 2.5 × 10 cm (Bio-Rad) was used to wash unbound proteins with 5 column volume (CV) of wash I buffer [50 mM HEPES pH 7.6, 150 mM NaCl, 50 mM imidazole pH 8.0, 1 mM TCEP, 5 % (v/v) glycerol, 0.1 % (w/v) lauryl maltose neopentyl glycol (LMNG), and with or without 0.02 % (w/v) CHS] and 12 CV of wash II buffer [50 mM HEPES pH 7.6, 150 mM NaCl, 50 mM imidazole pH 8.0, 1 mM TCEP, 5 % (v/v) glycerol, 0.01 % (w/v) LMNG, and with or without 0.002 % (w/v) CHS]. Bound proteins were eluted with elution buffer [50 mM HEPES pH 7.6, 150 mM NaCl, 250 mM imidazole pH 8.0, 1 mM TCEP, 5 % (v/v) glycerol, 0.005 % (w/v) LMNG, and with or without 0.001 % (w/v) CHS] and 0.5 mM EDTA was added to prevent aggregation. Amicon Ultra-15 100 kDa centrifugal filter unit (Merck Millipore) was used to enrich the eluate. Aggregates were removed by centrifugation at 16,000 $g$, 4 °C for 30 min. Recombinant proteins were separated on a Superdex 200 increase 10/300 GL column (GE Healthcare) equilibrated with final buffer [20 mM HEPES pH 7.6, 150 mM NaCl, 1 mM TCEP, 0.005 % (w/v) LMNG, with or without 0.001 % (w/v) CHS, and 0- or 10-mM malate pH 7.0]. Finally, proteins were concentrated at a final concentration of 2–3 mg mL$^{-1}$ using Amicon Ultra-0.5 100 kDa centrifugal filter unit (Merck Millipore).

### Cryo-electron microscopy sample preparation and data acquisition
For cryo-EM grid preparation, recombinant AtALMT9 samples were diluted to a final concentration of 1.7–1.9 mg mL$^{-1}$. Quantifoil R 1.2/1.3 Au 300 (Electron Microscopy Sciences) or Au-Flat R 1.2/1.3 Au 300 (Protochips) grids were negatively glow-discharged for 90 s with a PELCO easiGlow glow-discharge cleaning system (Ted Pella). Plunge freezing were conducted by a Vitrobot Mark IV (Thermo Fisher Scientific) with 3 μL of proteins, blotting time 2–2.5 s, blotting force 4–5, and 100% humidity at 4 °C at the Seoul National University Center for Macromolecular and Cell Imaging. For recombinant AtALMT1 sample, Quantifoil R 1.2/1.3 Au 200 (Electron Microscopy Sciences) were negatively glow-discharged for 60 s. Plunge freezing were conducted with blotting time 3.5 s and blotting force 5.

The dataset 1 (with CHS and 0 mM malate) was collected on a Glacios (Thermo Fisher Scientific) operated at an acceleration voltage at 200 kV with a Falcon 4 detector (Thermo Fisher Scientific) at the Seoul National University Center for Macromolecular and Cell Imaging. Movie data were collected using automated process of EPU software with electron counting mode. Total 11,640 movies were recorded and fractionated into 50 frames with total dose of 50 e$^-$ Å$^{-2}$, a pixel size of 0.87 Å per pixel, and defocus range of −0.7 to −1.9 μm. The dataset 2 (with CHS and 10 mM malate) and 3 (without CHS and 10 mM malate) were collected on a Titan Krios (Thermo Fisher Scientific) operated at an acceleration voltage at 300 kV with a K3 detector (Gatan) and a BioQuantum energy filter (Gatan) at the Institute of Membrane Proteins. Movie data were collected using automated process of EPU software with electron counting mode. Total 10,790 and 10,602 movies were recorded and fractionated into 50 frames with total dose of 60 and 50 e$^-$ Å$^{-2}$, a pixel size of 0.85 and 0.858 Å per pixel, and defocus range of −0.7 to −1.9 μm. The AtALMT1 dataset (with CHS and 0 mM malate) was collected on Talos Arctica (Thermo Fisher Scientific) operated at an acceleration voltage at 200 kV with a K3 detector (Gatan) and a BioQuantum energy filter (Gatan) at the Korea Basic Science Institute (KBSI). Total 4702 movies were recorded and

fractionated into 50 frames with total dose of 50 e$^-$ Å$^{-2}$, a pixel size of 0.84 Å per pixel, and defocus range of −0.7 to −1.9 μm.

## Image processing, model building and refinement

All image processing procedures were performed using cryoSPARC v4.0–4.4[62]. Dose-fractionated movies were processed by patch motion correction and patch contrast transfer function (CTF) estimation. Initial blob picking, template picking, and 2D classification were performed using a randomly selected small set of micrographs. Selected particles were used to construct initial ab initio model and to train Topaz[63] picking model. Particles extracted from the Topaz picking model were classified by 2D classification, ab initio reconstruction, and heterogeneous refinement. Topaz picking models were improved by these classified particles until there was no improvement. Using the final Topaz model, particles were extracted from whole dataset and classified through 2D classification and heterogeneous refinement with an ab initio model, decoys, and junks. Classified particles were refined through non-uniform refinement with C1 or C2 symmetry. CTF parameters and dose weighting were corrected by global CTF refinement and reference-based motion correction. Further 3D classification without alignment was used to separate heterogeneity around masked regions. In dataset 3, we were able to classify particles into many classes. However, many particles exhibited asymmetric features and classification results were inconsistent due to heterogeneity of the dataset 3. To obtain reliable classification results, we conducted symmetry expansion of consensus particles. Symmetry-expanded particles were used for 3D classification without alignment. To prove whether each class was reliable or not and to discriminate whether each particle was symmetric or not, we assumed that original particles with homogeneous conformation and their symmetry-expanded copies should be classified into (i) two identical, 180-degree rotational symmetric classes if the particles had asymmetric conformation, (ii) one class if they had symmetric conformation, and (iii) many classes for low-resolution particles or junks. To determine how many classes were needed for 3D classification, initial classifications were conducted by different class numbers. From many trials, we found out single classification for whole particles was not suitable for handling of heterogeneity from small differences such as lipid-bound states. Usually, many classes were mixed with others. Also, we realized that particles that distinct from all others tended to separate clearly according to our assumption. Therefore, we conducted many independent classifications for grouping classes sharing similar features. Then, we conducted subsequent classifications and tried to seek out reliable classes about lipid-bound states. We defined the reliable class based on the following rules: (1) particles should separate consistently from independent trials and different particle pools; (2) map should have clear and sharp side chain density around fenestration and pore; (3) all densities in pore and fenestration should be distinguishable, so that they can be modeled by known molecules of our expression and purification system; (4) particles must separate in accordance with our assumption. First of all, class N was separated from all others with two 180-degree rotational symmetric classes due to its highly asymmetric shape. Particles of symmetric classes were sorted into single symmetric class or multiple asymmetric classes, even though symmetry-expanded pairs are classified into same class. Symmetry-expanded pairs of low-resolution particles or junks were classified into the same classes. Thus, all symmetry-expanded pairs that classified into same classes were grouped into particle pool. Trans2 and cis2 classes were separated out from subsequent classifications. There was no difference in non-uniform refinement with C1 and C2 symmetry. We grouped remained asymmetric particles based on weak signal or strong signal in pore region. From weak signal group, class W was consistently classified and the rest featured non-clear, strong density in pore, so that the rest was added to the asymmetric group with strong signal in pore. Then, this group was subsequently classified into

three classes trans1, intermediate, and cis1-PI/PIP, and remainder that was difficult to identify. Most of classes were polished by subsequent 3D classification and refined by non-uniform refinement with C2 symmetry for symmetric classes and C1 symmetry (C2 symmetry relaxation) for asymmetric classes. For model building, maps were sharpened by local filtering. Detailed information of data processing is summarized in Supplementary Figs. 3–5. For initial model generation, the AF-Q9LS46-F1-model_v2 model from AlphaFold2 database[22,64] for AtALMT9 or previously reported structure (PDB: 7VQ4) for AtALMT1 was fitted into the local filtered map using UCSF ChimeraX[65]. Initial model was manually corrected by Coot[66] and automatically corrected by real space refinement of Phenix[67]. The final models of AtALMT9 contained all residues except flexible N-terminus, C-terminus, and two long loops (between H1 and H2 helices; between H5 and H6 helices). The final model of AtALMT1 contained all residues except flexible C-terminus and one long loop between H1 and H2 helices. The cryo-EM data collection, refinement, and validation statistics of all structures are summarized in Supplementary Table 1. All models and maps in the figures were visualized using UCSF ChimeraX[65].

## Vacuolar patch clamp recordings

All the constructs used were designed by us and mutagenesis done by Azenta life sciences on a pMDC83::ALMT9-GFP plasmid. For patch clamp experiments, *Nicotiana benthamiana* plants were grown for 4–6 weeks in 16 h light/ 8 h dark conditions. Transient overexpression of AtALMT9 and the mutants N142A; Y197A/S/F; T147A; R143K/Q; T205A; R200Q; R226K/Q; W120A/F were performed by agroinfiltration of leaves[68]. Leaves were harvested after 48 h, protoplasts were released by enzymatic digestion at 30 °C for 30–45 min. Enzymatic buffer contained 0.05% Pectolyase Y-23; 0.5% Cellulase R-10; 0.5% Macerozyme R-10; 1% BSA; 1 mM CaCl$_2$, 10 mM MES; pH 5.3 adjusted with KOH, osmolarity 500 mOsmol with sorbitol. After digestion, protoplasts were washed with a buffer containing 1 mM CaCl$_2$, 10 mM MES; pH 5.3 adjusted with KOH, and osmolarity 500 mOsmol with sorbitol. Vacuoles were released in the recording chamber by osmotic shock by adding 50 mM of EDTA. Expression levels of constructs were verified by fluorescence quantification in the tonoplast from vacuole confocal images (Supplementary Fig. 14). All enzymes were purchased from Duchefa. Chemicals purchased from Sigma Aldrich.

Patch-clamp experiments were performed with an epifluorescence microscope (Zeiss Axio Observer) equipped with a LED light source (CoolLED *p*E-300$^{ultra}$) to select vacuoles overexpressing AtALMT9-GFP and mutants. Glass pipettes (GC150T, Harvard Apparatus) had a resistance from 2.5 to 3.5 MΩ. Ionic currents were recorded using an EPC10 Patch clamp amplifier and the PatchMaster software (HEKA). Data analysis was performed using FitMaster (HEKA). Voltage stimulation protocol for current profile was made of a pre-pulse at +66 mV, and a 2.5 s voltage pulse from +66 mV to −114 mV in −20 mV decrements, the holding potential was 0 mV. Ramp protocol for voltage-current characteristic was made of a prepulse at +66 mV followed by a 3 s ramp from +66 to −154 mV. Currents were filtered at 1 kHz. The vacuolar side solution contained 11.2 mM malic acid; 100 mM HCl; pH 6 with Bis-Tris Propane (BTP), osmolarity 550 mOsmol with sorbitol. The cytosolic side solution contained 100 mM malic acid; 0.1 mM CaCl$_2$; pH 7.5 with BTP, osmolarity 500 mOsmol with sorbitol. Cytosolic buffer was exchanged using a gravity perfusion system coupled to a peristaltic pump (Ismatec Reglo ICC). In these conditions, the liquid junction potential was +6 mV and was corrected[69]. Only recordings with a resistance higher than 1 GΩ were used for the analysis. Activation kinetics were quantified using the FitMaster software (HEKA, Germany). The ion current traces from patches expressing AtALMT9 (WT and mutants) show an activation kinetics that can be described with a double exponential Eq. (1):

$$y(x) = Amp^0 + Amp^1*(1 - e^{(-x/tau1)}) + Amp^2*(1 - e^{(-x/tau2)}) \qquad (1)$$

$Amp^0$ is the initial amplitude value at the start of the current trace, $Amp^1$ the amplitude of the first exponential and $Amp^2$ at the end of the second exponential. All time constants, tau1 and tau2 were estimated in the range from the start of the −120 mV pulse until the plateau was reached, that is after 400 ms for *AtALMT9-R143Q* and 2500 ms for *AtALMT9*, *AtALMT9-R143K* and *AtALMT9-R200Q*. The first 2–6 ms after the start of the pulse were excluded to remove any residual capacitive current from the analysis. Statistical analysis were done using the non-parametric two-sided Mann-Whitney test.

## Confocal imaging of vacuoles

For confocal imaging, vacuoles of *N. benthamiana* plants were prepared following the same procedure as described in the "vacuolar patch clamp recordings" section. To identify the localization of the GFP for all tested constructs 12-bit images were acquired with a Leica STELLARIS 8 equipped with a 40x water immersion objective (HC PL APO CS2 40x/1.10 water). Fluorescent signal from the GFP and the chloroplasts were detected with the STELLARIS hybrid detectors, HyD X used in the counting mode. Emission was detected at 500–550 nm for the GFP and at 600–650 nm for chlorophyll auto-fluorescence upon excitation at 488 nm (White light laser). Transmitted light images were obtained with a PMT detector. The whole system was driven by LAS X software (Leica). Vacuole images were acquired in an image size of 1024*1024 px size format, with a pinhole of 1 airy and a zoom of 2 (pixel size: 142.05 nm). For GFP fluorescence quantification, images were acquired in a 512*512 px size format at a zoom of 2 with a pinhole of 3 airy (pixel size: 284.37 nm). Image analysis for fluorescence quantification at the vacuolar membrane was done using the ImageJ software. The background fluorescence was subtracted from images of GFP and chlorophyll. In order to quantify the fluorescence only in the vacuolar membrane we subtracted the chlorophyll fluorescence from GFP images. The vacuolar membrane region was selected manually.

## All-atom molecular dynamics simulations

The GROMACS 2024.1 package[70] was used to perform molecular dynamics (MD) simulations. The CHARMM36m and CHARMM36 force fields were employed for proteins[71] and lipids[72], respectively. Water molecules were described using the CHARMM-modified TIP3P model[71]. Parameters for malate molecules were generated using the CGenFF webserver[73]. The CUFIX corrections for charge-charge interactions were used to resolve the overestimated attractions between oppositely charged species[74]. Long-range electrostatic forces were computed using particle-mesh Ewald summation[75] with a 1.2 Å grid spacing and a 12 Å real space cutoff. Lennard-Jones forces were computed using a 10–12 Å switching scheme. The SETTLE[76] and LINCS[77] algorithms were used to constrain hydrogen atoms in water and non-water molecules, respectively. To set the temperature at 300 K during simulations, we employed the v-rescale scheme with a time constant of 1 picosecond. To set the pressure at 1 bar along the lateral and normal directions independently (i.e., zero surface tension), we used the c-rescale scheme with a semi-isotropic coupling option.

The AtALMT9 TMD (77–248) in the narrow or wide classes was embedded into a bilayer of POPC using the CHARMM-GUI webserver[78]. We used a hexagonal simulation box with dimensions of roughly 9 nm × 8 nm × 8.5 nm, where the angles α, β, and γ were set to 90°, 90°, and 60°, respectively. The complex systems were solvated with a solution including sodium, chloride, or malate. Each simulation setup was energy-minimized for 5000 steps. Subsequently, each system underwent a 250 ns equilibration at 1 bar and 300 K while applying position restraints to the protein backbone atoms. For the production runs to measure ionic currents, each system was simulated three times, each for three microseconds at 1 bar and 300 K under an external electric potential difference of zero or 500 mV along the z axis[79]. For all systems, changes in the positions of key arginine residues across three

replicate simulations converged reasonably well (Supplementary Fig. 12f). For ion transport analysis, we defined two planar boundaries parallel to the xy plane at distances of 1.5 nm from the protein's center of mass along the z axis. Ions crossing these boundaries were counted as transport events. Analysis of the production simulations was conducted using atomic coordinates saved every 100 picoseconds. All MD simulation visualizations were performed using VMD[80]. See Supplementary Table 2 for the list of simulation systems with varying ionic conditions and electric potential differences.

## Coarse-grained molecular dynamics simulations

Coarse-grained (CG) molecular dynamics simulations were performed using the Martini3 force field[54] to investigate the spatial distribution and conformational dynamics of lipids around the AtALMT9 TMD (77–248) in the narrow, wide, and cis2 classes over an extended time scale. All CG simulations were performed using the GROMACS 2024.1 package[70] with the simulation parameters consistent with the Martini3 such as the reaction field with a dielectric constant of 15 for electrostatics and the 1.1 nm cutoffs for Coulomb and Lennard-Jones interactions. The simulation systems were generated using the CHARMM-GUI Martini Bilayer Maker[81]. Each system contained a lipid bilayer of a single lipid type (DLPC, DOPC, DPPC, DBPC, or DGPC). The channel-lipid complex system was solvated using the Martini water model and $Na^+$ and $Cl^-$ ions corresponding to 150 mM. The dimensions of a rectangular simulation box were 8 nm × 8 nm × 10 nm, approximately. Each system was energy-minimized for 10,000 steps and equilibrated for 100 nanoseconds at 1 bar and 300 K. During the production runs at 1 bar and 300 K, trajectory data were saved every 10 nanoseconds. The protein beads were positionally restrained with a force constant of 1000 kJ mol$^{-1}$ nm$^{-2}$ during the simulations to maintain the overall structure. Lipid density map analyses were performed using the Densflux webserver[82]. See Supplementary Table 3 for the list of all simulation systems.

## Lipid analysis using LC-MS

For lipid extraction, purified proteins at 12.4 mg mL$^{-1}$ in 75 µL of sample buffer [20 mM HEPES pH 7.6, 150 mM NaCl, 1 mM TCEP, 0.005% (w/v) LMNG, and 10 mM malate pH 7.0] or the same volume of blank buffer were mixed with 300 µL of a cold chloroform:methanol mixture (2:1, v/v) according to the Folch's method[44]. The sample was vortexed for 1 min and centrifuged at 16,000 g for 20 min at 4 °C. The non-polar phase was transferred to a new Eppendorf tube® and dried under nitrogen gas. The lipid extract was then reconstituted in 80 µL of a chloroform:isopropanol:acetonitrile mixture (2.5:2:1:1, v/v/v/v). Lipid analysis was performed using an Agilent 1290 Infinity LC system (Agilent Technologies) coupled to a trapped ion mobility-quadrupole time-of-flight mass spectrometer (Bruker Daltonics). For separation, an ACQUITY Premier CSH C18 column with VanGuard FIT (2.1 mm × 100 mm, 1.7 µm; Waters) was used and maintained at 30 °C. The binary mobile phases comprised 10 mM ammonium acetate in water:acetonitrile (60:40, v/v, solvent A) and isopropanol:acetonitrile (90:10, v/v, solvent B). Samples were eluted at 0.2 mL min$^{-1}$ for 22 min. Gradient elution was conducted as follows: 40–55% B from 0 to 6 min, 55–60% B from 6 to 11 min, 60 to 99% B from 11 to 14 min, 99% B from 14 to 18 min, 99 to 40% B from 18 to 18.1 min, and 40% B from 18.1 to 22 min. Lipid identification was carried out by comparing the experimental data with MS2 spectra from LIPID MAPS, HMDB, MS DIAL.msp library, and an in-house library.

## Reporting summary

Further information on research design is available in the Nature Portfolio Reporting Summary linked to this article.

## Data availability

The data that support this study are available from the corresponding authors upon request. Coordinates have been deposited in the Protein

Data Bank under accession codes 8ZTE (AtALMT9 sterol1 class), 8ZTG (AtALMT9 sterol2 class), 8ZTH (AtALMT9 N class), 8ZTI (AtALMT9 W class), 8ZTJ (AtALMT9 cis1-PI/PIP class), 8ZTK (AtALMT9 cis2 class), 8ZTL (AtALMT9 intermediate class), 8ZTM (AtALMT9 trans1 class), 8ZTN (AtALMT9 trans2 class), and 9JTW (AtALMT1). The corresponding cryo-EM density maps with local-filtering, half-maps and masks have been deposited in the Electron Microscopy Data Bank under accession codes EMDB-60459 [https://www.ebi.ac.uk/pdbe/entry/emdb/EMD-60459] (AtALMT9 sterol1 class), EMDB-60461 [https://www.ebi.ac.uk/pdbe/entry/emdb/EMD-60461] (AtALMT9 sterol2 class), EMDB-60462 [https://www.ebi.ac.uk/pdbe/entry/emdb/EMD-60462] (AtALMT9 N class), EMDB-60463 [https://www.ebi.ac.uk/pdbe/entry/emdb/EMD-60463] (AtALMT9 W class), EMDB-60464 [https://www.ebi.ac.uk/pdbe/entry/emdb/EMD-60464] (AtALMT9 cis1-PI/PIP class), EMDB-60465 [https://www.ebi.ac.uk/pdbe/entry/emdb/EMD-60465] (AtALMT9 cis2 class), EMDB-60466 [https://www.ebi.ac.uk/pdbe/entry/emdb/EMD-60466] (AtALMT9 intermediate class), EMDB-60467 [https://www.ebi.ac.uk/pdbe/entry/emdb/EMD-60467] (AtALMT9 trans1 class), EMDB-60468 [https://www.ebi.ac.uk/pdbe/entry/emdb/EMD-60468] (AtALMT9 trans2 class), and EMDB-61818 [https://www.ebi.ac.uk/pdbe/entry/emdb/EMD-61818] (AtALMT1). The molecular dynamics simulation files, including trajectories, structures, and parameters, have been deposited in the Zenodo under accession code 14177701. The mass spectrometry data have been deposited in the ProteomeXchange Consortium via the PRIDE partner repository under accession code PXD059651. The referenced coordinates have been deposited in the Protein Data Bank under accession codes 7W6K (GmALMT12), 7VQ4 (AtALMT1 apo / pH 7.5), 7VQ7 (AtALMT1 Al$^{3+}$ / pH 5.0), 8HIW (AtALMT9 apo / pH 7.5) 8HIY (AtALMT9 malate / pH 7.5), and 8ZVF (AtALMT9 malate/pH 5.0). The source data underlying Figs. 1g–i, 2j, k, 3b, d, e, 4c, i, j, l, m, and 5h, Supplementary Figs. 1b, 8, 10b–g, 11b, d, f, h, j, 12d–f, 13a, c–f, h, i, 14b, and 20e are provided as a Source Data file. Source data are provided with this paper.

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

## Acknowledgements

Cryo-EM data collection was performed in part at the Korea Basic Science Institute (Leading Research Equipment User Program, C230430). Computational and network resources were provided in part by KREONET and GSDC, KISTI. Supercomputing resources including technical support was granted by the National Supercomputing Center (KSC-2023-CRE-0341). We thank C. Corratgé-Faillie at the Plant Electrophysiology Platform (PEP) from the Institute for Plant Sciences of Montpellier (IPSiM) for technical support and assistance. We also thank C. Alcon at the imaging facility MRI, a member of the France-BioImaging infrastructure and supported by the French National Research Agency (ANR-10-INBS-04), for assistance. S.L. was supported by the Basic Science Research Program (RS-2022-NR069539) and the Bio & Medical Technology Development Program (RS-2024-00440289, RS-2023-00223552 and RS-2021-NR056573) through the National Research Foundation of Korea (NRF) grants. A.D.A. was supported by the French National Research Agency ANR (project Netflux) and by CNRS ATIP-AVENIR 2018. J.Y. was supported by an NRF grant (2020R1A2C1101424) and an Institute of Information & communications Technology Planning & Evaluation (IITP) grant (RS-2021-II212068). G.S.H. was supported by a grant from Korea Basic Science Institute (A423200).

## Author contributions

Y.L. prepared samples and performed structural analysis. E.D.C. and J.J. performed patch-clamp experiments, and E.D.C. and A.D.A. analyzed electrophysiologial data. S.J. prepared samples. J.M.Y. and J.Y. performed MD simulation analysis. S.Y.J. and G.S.H. performed mass spectrometric analysis. Y.L., A.D.A., and S.L. wrote the manuscript with help from all authors. A.D.A. directed and supervised the electrophysiology research. S.L. directed and supervised all of the research.

## Competing interests

The authors declare no competing interests.
