## [Transparent Peer Review file · Nature Communications]

Structural basis for malate-driven, pore lipid-regulated activation of the Arabidopsis vacuolar anion channel ALMT9

Corresponding Author: Professor Sangho Lee

Version 0:

Reviewer comments:

Reviewer #1

(Remarks to the Author)

Lee et al. generated cryo-EM structure of ALMT9 from Arabidopsis thaliana (AtALMT9), a malate-activated vacuolar anion channel in various states. In all the states, lipids were found to interact with the ion conduction pore of the channel. Based on patch clamp studies and molecular dynamics simulations the authors propose a voltage dependent activation mechanism based on the competition between pore-lipids and malate at the entrance to ALMT9's ion conductive pathway.

To validate models, derive from structural analysis of frozen ion channel and solute transporter proteins by electrophysiological analysis is the way to go to learn about the time-dependent activities of these membrane intrinsic moieties.

The authors have provided a wonderful example of how one can find out about the structure and performance of ion channels located in the vacuole, central organelle of the plant cell.

The authors study on ALMT9 breaks new ground in understanding the structure-function relationship of plant ion channels. For the reader to better understand the issue the authors may wish to respond to the comments and questions listed below.

Comments and Questions:

To test this hypothesis based on cryo-EM work, the De Angeli lab performed electrophysiological experiments on tobacco vacuoles over-expressing Arabidopsis ALMT9 wildtype WT and mutants MTs.

Comment: Kovermann et al. 2007 showed via patch-clamp analysis that an Atalmt9 T-DNA insertion mutant in Arabidopsis exhibited a 70% reduced vacuolar malate channel activity. Thus, questions to the nature of the remaining 30% to the natural home of AtALMT9.

Question 1: Although the tobacco transient expression system has been used routinely, the reader may want to know, how much of the tobacco intrinsic ALMT9 and other vacuolar ALMTs contribute to the currents measured.

Was the expression rate of tobacco intrinsic and Arabidopsis ALMT6 monitored by e.g. qPCR analysis?

Another way to answer this question is to transiently transform protoplast of the Arabidopsis almt9 loss-of-function mutant with the key constructs used in the tobacco system and patch clamp the vacuoles isolated thereof.

Question 2: Where WT and MT ALMT9's expressed to the same extent and targeted to the vacuole membrane? For targeting issues in 2002 Kovermann et al. used AtALMT9-GFP constructs. Now one would e.g. use GFP antibodies to quantify the channel protein density and localization. What about the ALMT9 current amplitude and voltage dependence and kinetics, when the authors mutated the neighbors to the key residues pinpointed by cryo-EM?

Two residue N142 and Y197 were spotted as major sites in the interaction between the channel-pore structural lipid. In vacuoles expressing the MT N142A malate ionic currents were found dramatically decreased when compared to WT. Y197A, Y197S and Y197F mutants were behaving in a similar manner as N142A and lacked malate activation in presence of chloride. These data would be consistent with the role of the N142-Y197 interaction in stabilizing the wide-like conformation, which is permeable to anions.

Question 1: for position Y197 3 mutations were analyzed via patch clamp studies. Did the authors as well tested mutants in residue N142 other than T>A and what was the trend in ALMT9 ionic currents amplitude and malate effect looked like?

Question 2: The same questions applies to the functional validation of the cryo-EM spotted threonine residue mutant T147A.

The authors found that patches expressing T147A displayed malate currents that were significantly reduced compared to

WT (Fig. 4j). And claim these data demonstrated that T147 plays an essential role in AtALMT9 and, together with the structural data, highlighted the importance of the fenestration in the functioning of AtALMT9.

Question: The authors in the Introduction propose that voltage-dependent and slow activation kinetics Vacuolar ALMTs including AtALMT9 present might involve uncharted modulatory elements such as membrane lipids. Compared to T147A, however, the current reduction in the mutants assumed to be involved in fenestration is much smaller. So, what about the authors hypothesis that mutations addressing the lipid fenestration of AtALMT9 will affect the activation kinetics?

Authors statement: Our cryo-EM structures, obtained at zero membrane potential, presumably reflect a non-conductive, basal conformational ensemble at near-zero tonoplast potential such as those found in closed stomata (Ref.: 9 Kovermann et al. 2007).

MD simulations demonstrate that arginine residues in the pore undergo voltage-dependent conformational changes.

Comment: Reference 9 Kovermann et al. 2007 was cited for guard cells vacuole membrane potential but this citation is wrong. The guard cell vacuole potential has not been measured so far, but that of root hairs (Wang et al. 2015). In the latter system the membrane potential is around -30 mV when hyperpolarized and around zero when depolarized.

Question: What can the authors, when taking this membrane potential range and physiological ion gradients into account, MD simulations will tell the reader about the mode of ALMT9 action?

Authors write: .. it was reported that the glutamate and asparagine substitutions R143E, R143N, R226E, and R226N in AtALMT9 completely abolish ion currents compared to WT49. To investigate further whether the charge and the shape of side chain could influence AtALMT9 functioning given the voltage-dependent movements of R143 in the MD simulations, we performed vacuolar patch clamp recordings expressing the R143K and R143Q mutants (Fig. 5f-h).

Data show, however, that the long side chain length and a non-negative charge are the minimal requirements to guarantee anion transport.

Question: What is the take-home message for the reader ?

Authors write: Taken together, our results from MD simulations and vacuolar patch clamp recordings corroborate the structural roles of R143, R203 and R226 in activation of ALMT9.

Comment: Given that beside T143A all other mutations altered the properties of ALMT9 only marginally, the authors may want to tone down this statement.

Authors chapter: Conservation of key structural determinants in ALMTs

Comment: I read this rather long chapter several times but did not find the take-home message. When there is no story to tell the reader, this chapter should be condensed to a paragraph, like that in the Discussion: "Interestingly, sequence alignment suggest that this stepping hydrophilic pathway is conserved only in the clade 2 ALMTs (Fig. 6). These results could suggest that plugging/unplugging the pore from lipids may be involved in the regulation of ALMT9 ion transport activity and possibly in other ALMTs of the clade 2"

Authors write in the DISCUSSION, starting from: The clade 2 ALMTs constitute ..

Comment: the first paragraph is redundant and can be omitted.

Authors write: The clade 2 ALMTs constitute important anion channels responsible for transporting malate and chloride across the vacuolar membrane of guard cells which is essential for the regulation of stomata aperture.

Comment: In the Introduction and Discussion guard cells and the regulation of stomata aperture are mentioned. However, there is not a single result, that is gained with vacuoles from stomatal guard cells. There is also no experiment showing stomatal movement of guard cells expressing any of the mutants used in this study. Thus, mentioning open and closed stomata in the cartoon Figure 7 seems not justified.

Question: Given that Kovermann et al. 2007 showed that ALMT9 is predominately expressed in mesophyll cell, the reader in the Introduction and/or Discussion would like to learn what the physiological role of malate induced vacuolar chloride fluxes are. Do we know about stimulus induced transients in cytosolic malate and associated chloride fluxes across the vacuole membrane?

Reviewer #2

(Remarks to the Author)

ALMTs form a family of anion channel in plants and play important roles in the regulation plant anion homeostasis. Previous studies on AtALMT1 and GmALMT12 reveal overall architectures, anion selectivity, and gating mechanisms of ALMTs. In this study, the authors present structures of another ALMT protein AtALMT9 in different conformations, along with electrophysiological assays and computational studies. This manuscript is lengthy and inconclusive, and should be substantially revised before publication.

1. Although the overall map quality is good, the identification of lipids, malate, sterols, and cis1-PI4P solely based on the ligand density is questionable. For example, where did this cis1-PI4P come from? How did the authors distinguish it from POPC? Without more evidence, the authors can not accurately assign these ligands, especially for malate.

2. The sterol mimic-bound and lipid-bound AtALMT9 structures were overinterpreted. Clearly, these structures are in non-conducting conformations, which were probably induced during the protein sample preparation. To me, these structures, obtained from insect cells but used for the analysis of lipid-protein interactions for a plant ion channel, provide limited information for the channel activation. The authors should focus on the other two conformations.

(Remarks to the Author)

In the manuscript by Lee et al., the authors have resolved the structure of AtALMT9 in various lipid-bound states. Their study reveals that lipid tail length regulates the pore width, thereby allowing anion permeation. This research is of broad interest to the community working on channel proteins, particularly in understanding how lipids regulate gating mechanisms. However, I have a few concerns regarding the molecular dynamics (MD) simulations that the authors should address.

Major Comments:

1) Pressure Coupling in MD Simulations: The authors have turned off pressure coupling during their MD simulations. This choice can lead to unrealistic system behavior, especially in terms of membrane properties, protein-lipid interactions, and the overall physical realism of the simulation. Lipid dynamics within the bilayer, including flip-flop and lateral diffusion, are pressure-dependent. Turning off pressure coupling can result in non-physiological lipid behavior, which in turn affects membrane dynamics and the function of embedded proteins. Given that protein-lipid interactions are critical in this study, as they influence the pore widths, it is concerning that pressure coupling was not employed. Notably, several MD simulations studying lipid pore formation under external electric fields, even up to 600 mV, have been successfully conducted with pressure coupling turned on, yielding stable systems across various lipid types and electric fields (Reference: <https://doi.org/10.1021/jacs.2c08543>). I strongly recommend that the authors consider repeating their simulations with pressure coupling enabled.

2) Lipid Density Map Analysis: In lines 393-395, the authors mention that they set up MD simulations by replacing peripheral lipids modeled based on cryo-EM structures with POPC lipids. However, they have not provided a lipid density map analysis to confirm whether these lipids continue to occupy the cryo-EM observed binding sites during the MD simulations. This analysis is crucial to ensure the accuracy of the modeled interactions.

3) Lipid Tail Insertion and Stability: In lines 82-84, the authors discuss how the degree of lipid tail insertion from peripheral lipids regulates pore widths. Although they suggest that the tail lengths are likely 16 or 18 carbon atoms based on insect cell lipid composition (lines 129-133), they also acknowledge a small percentage of lipids with 20 carbon atoms. Longer lipid tails and unsaturation provide the structural flexibility needed to bind specific amino acids with their headgroups while positioning hydrophobic tails in the voids between helices. Given the flexible nature of lipid tails, it is challenging to confirm from the tube-like densities observed in cryo-EM whether they correspond to 16 or 18 carbon atoms. Moreover, the structural arrangement of lipid tails (C16/C18/C8/C10) shown in Figure 2 raises questions about their energetic stability. I suggest the authors conduct coarse-grained simulations with varying membrane compositions in tail length and degree of unsaturation. These simulations are relatively inexpensive and could help clarify which lipid tail configurations are energetically stable and best fit the tube-like densities observed in the cryo-EM structure.

4) Fatty Acid Orientation in Figure 2a: In Figure 2a, the fatty acid tail is oriented such that the carboxyl group faces the hydrophobic bulk lipids, while the hydrophobic tail faces the pore. Have the authors considered rotating the fatty acid tail so that the carboxyl group faces the pore, potentially forming water-mediated electrostatic interactions with R143up?

5) Fatty Acid Orientation in Figure 2b: Similarly, in Figure 2b, the peripheral fatty acids FA1, FA2, FB1, and FB2 have their carboxyl groups facing the hydrophobic bulk lipids. Rotating the lipids would position the headgroups towards the pore, with the two acyl chains occupying the two tube-like densities. Have the authors considered this alternative lipid orientation, and would it remain stable at the binding site?

Minor Comments:

1) MD Simulation Methods Section: The MD simulation methods section lacks sufficient detail for reproducibility. For instance, the authors refer to results using structures representing unplugged states (Fig 5a, e and SI figures and movies), but the specific unplugged structures used in the MD simulations are not clearly referenced (e.g., Fig 2a, b). The authors should clarify the use of unplugged (N and W states) structures in the MD methods section and mention them in the SI Table 2 heading.

2) Protonation States and Thermostat Parameters: The methods section does not explain how the protonation states of the amino acids were determined, nor does it provide information on the total charge of the protein in different states or the input parameters used for the thermostat.

3) Terminology Clarification in Lines 89-90: The authors could improve clarity by explicitly referring to the "peripheral lipid-only class" as "unplugged states."

4) Movie Clarity: The movies provided by the authors are very clear. However, it would be beneficial to also include the peripheral lipids in the visualizations.

5) Data Availability: A link to a public repository containing the simulated structures and topology files is missing. Additionally, structures with PIP and cholesterol bound are absent. Uploading these structures to a public repository would significantly benefit the research community.

Version 1:

Reviewer comments:

Reviewer #1

(Remarks to the Author)

Structural information on plant ion channel are quite rare and besides TPC1 vacuole ion channel 3D structures are even more so.

This state-of-art Cryo-EM study should be published.

The revised version of the paper has improved much concerning the story board and proper controls.

Reviewer #2

(Remarks to the Author)

I am impressed by the authors' efforts in addressing my major concerns, as well as others'. In my opinion, this story presents too much experiment data from structures, EPs, to MD simulations. Although I do not totally agree with the authors' conclusion, given the other ALMT9 story, whose structures are in low resolutions, was published recently, in principle I would recommend the publication of this manuscript. Nevertheless, I wish the authors seriously consider my comments. The structures presented in this study are convincing and cross-validated by the other study. However, the interpretation should be cautious. It is not unusual to observe lipids in the ion conducting pathway in ion channel structures, but not all these lipids are functional or physiology-relevant. To me, LMNG is not a perfect detergent and sometimes induces artifacts. As a control, the authors use LMNG to determine an ALMT1 structure. Why didn't they use a different detergent, for example, GDN, to determine the same ALMT9 as a control, which is more straightforward and relevant?

Reviewer #3

(Remarks to the Author)

After carefully reviewing the revised manuscript and the authors' responses to my previous comments, I am satisfied that all my concerns have been addressed. The authors have provided clear and thorough explanations, improving the overall quality and clarity of the manuscript.

I am pleased to recommend this manuscript for publication.

Minor Comment:

Authors could include a brief description in the Methods section explaining how chloride permeation events and lipid density maps were analyzed. This addition would enhance the reproducibility of their work for readers.

Overall, I commend the authors for their effort in addressing the feedback and presenting a well-prepared manuscript.

POINT-BY-POINT RESPONSES

We sincerely appreciate the constructive comments and critiques which helped improve our study. We present summary followed by detailed point-by-point responses.

Summary

In the revised manuscript, we conducted the following experiments to answer the questions that the reviewers raised.

Reviewer #1:

- Electrophysiological recordings on more site directed mutants: Fig. 4k-m, Fig. 5g, h, Supplementary Fig. 10, 13
- Confocal imaging to verify the expression levels of all mutants: Supplementary Fig. 14

Reviewer #2:

- Mass spectrometric analysis to specify lipid species present in the sample: Fig. 1g-i, Supplementary Fig. 8
- Structure determination of AtALMT1 to address concerns regarding potential detergent-induced conformational changes: Supplementary Fig. 21, Supplementary Table 1

Reviewer #3:

- All-atom and coarse-grained molecular dynamics (MD) simulations in accordance with the reviewer's advice: Fig. 4f-j, 5i, Supplementary Fig. 12, 19, Supplementary Table 2, 3, Supplementary Movie 1-12

From these results, we revised and reorganized the manuscript to convey the new findings better, as outlined below.

1. We added the new Fig. 1g-i, Supplementary Fig. 8 and a paragraph describing the lipids identified from the mass spectrometry.
2. We rearranged the original Fig. 2j to the new Fig. 2j-k with addition of a negative control with empty vector.
3. We rearranged and revised the Figures and Results related to the original Fig. 4-5 to concentrate sequentially on the ion conduction and lipid migration. Specifically, we moved the original Fig. 4f-h to the new Fig. 5a-c; the original Fig. 4i-j to the new Fig. 5g-h; the original Fig. 5a-e to the new Fig. 4f-j with new simulation results, better description, and arrangement; the original Fig. 5f-h to the new Fig. 4k-m with new data from more mutants. We added the new Fig. 5i to show coarse-grained simulation results.
4. To better focus on the mechanism of AtALMT9, we relocated the original Fig. 6 and its description about the structural comparisons to Supplementary Fig. 16. Subsequently, the original Fig. 7 to show a proposed mechanism has been changed to the new Fig. 6 with improved labels.
5. We reorganized Supplementary Information as a result of adding new data.

- A. We moved the original Supplementary Fig. 8 to the new Supplementary Fig. 9 due to the addition of the Supplementary Fig. 8 showing mass spectrometry data; the original Supplementary Fig. 9 to the new Supplementary Fig. 11 due to the addition of the new Supplementary Fig. 10 showing electrophysiological data for W120A/F mutants; the original Supplementary Fig. 10 to the new Supplementary Fig. 15; the original Supplementary Fig. 12 to the new and expanded Supplementary Fig. 13 with new data from R200Q and R226K/Q mutant; the original Supplementary Fig. 13 to the new Supplementary Fig. 17; the original Supplementary Fig. 14 to the new Supplementary Fig. 18; the original Supplementary Fig. 15 to the new Supplementary Fig. 20.
 - B. We added the following supplementary figures: the new Supplementary Fig. 8 for mass spectrometry; the new Supplementary Fig. 14 showing expression levels of all mutants used for electrophysiology experiments; the new Supplementary Fig. 19 showing data from coarse-grained simulations; the new Supplementary Fig. 21 showing cryo-EM analysis of Arabidopsis ALMT1; the new Supplementary Table 3 showing coarse-grained simulation system summary.
 - C. We expanded and revised the original Supplementary Fig. 11 to the new Supplementary Fig. 12 to include new all-atom simulations at zero membrane potential and statistics; the original Supplementary Movie 1-5 to the new Supplementary Movie 1-12 showing new MD simulation results; the new Supplementary Table 1-2 to incorporate new data from cryo-EM analysis of AtALMT1 and all-atom simulations.
6. We removed redundant paragraphs in the Discussion and added a new paragraph addressing the limitations of the study.
 7. We revised many sentences within the paragraphs and adjusted several terms to enhance clarity and align with the new findings.
 8. We also revised the Abstract, Introduction, Methods, and other sections to reflect these changes.

Reviewer #1

Lee et al. generated cryo-EM structure of ALMT9 from *Arabidopsis thaliana* (AtALMT9), a malate-activated vacuolar anion channel in various states. In all the states, lipids were found to interact with the ion conduction pore of the channel. Based on patch clamp studies and molecular dynamics simulations the authors propose a voltage dependent activation mechanism based on the competition between pore-lipids and malate at the entrance to ALMT9's ion conductive pathway.

To validate models, derive from structural analysis of frozen ion channel and solute transporter proteins by electrophysiological analysis is the way to go to learn about the time-dependent activities of these membrane intrinsic moieties.

The authors have provided a wonderful example of how one can find out about the structure and performance of ion channels located in the vacuole, central organelle of the plant cell.

The authors study on ALMT9 breaks new ground in understanding the structure-function relationship of plant ion channels. For the reader to better understand the issue the authors may wish to respond to the comments and questions listed below.

Comments and Questions:

To test this hypothesis based on cryo-EM work, the De Angeli lab performed electrophysiological experiments on tobacco vacuoles over-expressing *Arabidopsis* ALMT9 wildtype WT and mutants MTs.

Comment: Kovermann et al. 2007 showed via patch-clamp analysis that an *Atalmt9* T-DNA insertion mutant in *Arabidopsis* exhibited a 70% reduced vacuolar malate channel activity. Thus, questions to the nature of the remaining 30% to the natural home of AtALMT9.

The study that the reviewer mentioned (Kovermann et al. 2007) reported that in *almt9-1* knock-out (KO) that malate currents were 70 % reduced. The nature of the remaining 30 % currents still present in the KO vacuoles was not known. A recent study (Doireau, R., et al. 2024) shows that other ALMT channels like ALMT5 mediate organic acid currents across *Arabidopsis* vacuoles. These organic acid currents are likely to be responsible of the remaining 30%. We are currently conducting experimental work to address these issues. However, even if we agree that this is interesting we would point that it is out of the scope of the current manuscript where we aimed at solving the structural basis of the functional properties of AtALMT9.

Question 1: Although the tobacco transient expression system has been used routinely, the reader may want to know, how much of the tobacco intrinsic ALMT9 and other vacuolar ALMTs contribute to the currents measured.

Was the expression rate of tobacco intrinsic and *Arabidopsis* ALMT6 monitored by e.g. qPCR analysis?

Another way to answer this question is to transiently transform protoplast of the *Arabidopsis* *almt9* loss-of-function mutant with the key constructs used in the tobacco system and patch clamp the vacuoles isolated thereof.

Response: As the reviewer says it is indeed important to understand the potential contribution of endogenous currents that could be present in tobacco. We actually have addressed this question in two ways. We performed patch clamp analysis of vacuolar cytosolic side out

patches obtained from protoplasts extracted from tobacco plants transformed with empty vector only and we systematically compared ion currents from these patches with those from vacuoles expressing *AtALMT9* wild type (WT) and mutants. *AtALMT9* WT expressing patches displayed ion currents with voltage-dependent activation that were not detected in empty vector transformed patches. These currents were 5.5 times higher than empty vector background ion currents (Supplementary Fig. 1). Further, the expression of some of non-functional *AtALMT9* mutants such as N142A and R226Q showed the currents similar to those observed in empty vectors. These data therefore demonstrate that the presence of eventual endogenous currents that would originate from tobacco ALMTs do not interfere with *AtALMT9*-mediated currents. We have revised the manuscript and figures (i.e., Fig. 4k, m) to make these points explicit:

[Lines 268-269 of the revised manuscript]

“The currents in patches over-expressing N142A were the same as those recorded in the empty vector transformed vacuoles (Fig. 2i-k, Supplementary Fig. 1).”

[Lines 412-413 of the revised manuscript]

“Notably, unlike R143 and R200, R226K/Q mutants were unable to conduct malate currents (Fig. 4k, m, Supplementary Fig. 13e).”

Question 2: Where WT and MT ALMT9's expressed to the same extent and targeted to the vacuole membrane? For targeting issues in 2002 Kovermann et al. used *AtALMT9*-GFP constructs. Now one would e.g. use GFP antibodies to quantify the channel protein density and localization. What about the ALMT9 current amplitude and voltage dependence and kinetics, when the authors mutated the neighbors to the key residues pinpointed by cryo-EM?

Response: To ensure that the mutants were expressed in tobacco vacuoles, we performed patch clamp experiments expressing an *AtALMT9*-GFP fusion. We employed fluorescence microscopy to ascertain that we only patched vacuoles expressing the *AtALMT9*-GFP variants. We apologies if this was not clear in the original manuscript. We made it clearer in the revised manuscript (Supplementary Fig. 14a).

Further, to investigate if the mutants and the WT ALMT9-GFP were expressed at similar levels in the tonoplast, we used quantitative confocal microscopy on isolated vacuolar membrane (Supplementary Fig. 14b and Material and Methods). For this, we expressed the different ALMT9-GFP versions, isolated vacuoles and quantified the fluorescence in the vacuolar membrane only. We chose this method to quantify expression since the relevant quantification was to quantify the different versions of *AtALMT9*-GFP present in the tonoplast. Indeed, patch clamp data depends only on the proteins present in the isolated tonoplast. The data we obtained show that all the mutants we tested presented the same tonoplastic expression levels as the WT even when the mutants were not functionally active. We revised the manuscript as follows to reflect this point:

[Lines 769-770 of the revised manuscript]

“All the constructs used were designed by us and mutagenesis done by Azenta life sciences on a pMDC83::*ALMT9*-GFP plasmid.”

[Lines 784-786 of the revised manuscript]

“Patch-clamp experiments were performed with an epifluorescence microscope (Zeiss Axio Observer) equipped with a LED light source (CoolLED pE-300^{ultra}) to select vacuoles overexpressing AtALMT9-GFP and mutants.”

Two residue N142 and Y197 were spotted as major sites in the interaction between the channel-pore structural lipid. In vacuoles expressing the MT N142A malate ionic currents were found dramatically decreased when compared to WT. Y197A, Y197S and Y197F mutants were behaving in a similar manner as N142A and lacked malate activation in presence of chloride. These data would be consistent with the role of the N142-Y197 interaction in stabilizing the wide-like conformation, which is permeable to anions.

Question 1: for position Y197 3 mutations were analyzed via patch clamp studies. Did the authors as well tested mutants in residue N142 other than T>A and what was the trend in ALMT9 ionic currents amplitude and malate effect looked like?

Question 2: The same questions apply to the functional validation of the cryo-EM spotted threonine residue mutant T147A.

Response: The N142A abolishes the function of AtALMT9 like Y197A. All the changes we made in Y197 yielded the same effect on AtALMT9 transport capacities. Indeed, removing the hydroxyl group in Y197F, which disrupts the hydrogen bond, is sufficient to make AtALMT9 not functional. Further, the Y197S data shows that the length is also crucial for the interaction. Therefore, we do not think that testing more mutants of N142 would bring more information.

Regarding T147A, we decided to test the T205A mutant rather than to test a new variant of T147 because the structural analysis revealed that T205 is also a part of the lateral fenestration. In this way, we could obtain a more comprehensive view on the functional roles of residues in the lateral fenestration.

The authors found that patches expressing T147A displayed malate currents that were significantly reduced compared to WT (Fig. 4j). And claim these data demonstrated that T147 plays an essential role in AtALMT9 and, together with the structural data, highlighted the importance of the fenestration in the functioning of AtALMT9.

Question: The authors in the Introduction propose that voltage-dependent and slow activation kinetics vacuolar ALMTs including AtALMT9 present might involve uncharted modulatory elements such as membrane lipids. Compared to T147A, however, the current reduction in the mutants assumed to be involved in fenestration is much smaller. So, what about the authors hypothesis that mutations addressing the lipid fenestration of AtALMT9 will affect the activation kinetics?

Response: We also tested an alanine substitution of T205 (another conserved threonine), which resulted in a less pronounced but still significant reduction in malate currents. We revised the manuscript as follows to reflect this point:

[Lines 478-486 of the revised manuscript]

“To validate the functional roles of the threonine residues constituting the hydrophilic patches in fenestration, we prepared T147A and T205A mutants and analyzed its functionality by electrophysiology (Fig. 5g, h). We found that patches expressing T147A displayed abolished malate currents and that T205A showed significantly rescued malate currents (Fig. 5g, h). T147A and T205A seem to disrupt the hydrophilic connection between

the lipid head group binding site on the pore and the hydrophilic paths located outside of the fenestration (Fig. 5b). These data support the role of T147 and T205 that play an essential function in AtALMT9 and, together with the structural data, highlight the importance of the hydrophilic fenestration in the functioning of AtALMT9.”

Authors statement: Our cryo-EM structures, obtained at zero membrane potential, presumably reflect a non-conductive, basal conformational ensemble at near-zero tonoplast potential such as those found in closed stomata (Ref.: 9 Kovermann et al. 2007).

MD simulations demonstrate that arginine residues in the pore undergo voltage-dependent conformational changes.

Comment: Reference 9 Kovermann et al. 2007 was cited for guard cells vacuole membrane potential but this citation is wrong. The guard cell vacuole potential has not been measured so far, but that of root hairs (Wang et al. 2015). In the latter system the membrane potential is around -30 mV when hyperpolarized and around zero when depolarized.

Question: What can the authors, when taking this membrane potential range and physiological ion gradients into account, MD simulations will tell the reader about the mode of ALMT9 action?

Response: To observe sufficient ion flux during the limited time interval during which the MD simulations were performed, we used a higher electric field (-500 mV) than physiological membrane potential to accelerate the simulations. This strategy was already successful in simulating voltage-dependent activation of various ion channels (Jensen, M. Ø., et al. 2012; Kopec, W., et al. 2018; Kasimova, M. A., et al. 2019; and Liu, C., et al. 2023). Regardless of the physiological ranges of membrane potential, the simulations in this study provide insights into the conformational transition and ion conduction under a cytosolic negative potential.

Authors write: ... it was reported that the glutamate and asparagine substitutions R143E, R143N, R226E, and R226N in AtALMT9 completely abolish ion currents compared to WT⁴⁹. To investigate further whether the charge and the shape of side chain could influence AtALMT9 functioning given the voltage-dependent movements of R143 in the MD simulations, we performed vacuolar patch clamp recordings expressing the R143K and R143Q mutants (Fig. 5f-h).

Data show, however, that the long side chain length and a non-negative charge are the minimal requirements to guarantee anion transport.

Question: What is the take-home message for the reader?

Response: Taking the comprehensive results, we concluded that R143 and R226 are crucial for ion conduction and the voltage-dependent channel activation. Indeed, when R143 is mutated in a Q the voltage dependency is shifted and the activation kinetics becomes significantly faster. In the case of R226 this residue is strictly required since even the mutation in a K completely disrupts the transport capacity. The MD simulations reveal that R226 and R143 undergo voltage-dependent movements. Our results from structural analysis, electrophysiological studies using structure-based mutants and simulations enable us to propose a plausible mechanism for voltage-dependent ion transport in AtALMT9. We revised the manuscript as follows to highlight this point:

[Lines 396-416 of the revised manuscript]

“Our present findings indicate that R143, R200, and R226 are important for anion permeation. It has been reported that R143E/N, R200E, and R226E/N completely abolish ion currents compared to WT, except R200K/N⁴⁹. Such observations suggest the importance of the charge and size of these residues in ion transport by AtALMT9. Subsequently, we performed vacuolar patch clamp recordings expressing R143K/Q, R200Q, and R226K/Q mutants (Fig. 4k-m, Supplementary Fig. 13). The glutamine substitution removes the positive charge but can interact with malate, and features a side chain longer than asparagine. R143K mutant, keeping the positive charge but possessing a different shape, behaved similarly to WT (Fig. 4k-m, Supplementary Fig. 13a-d). R143Q mutant, which has no positive charge and a longer side-chain compared to asparagine, showed malate currents similar to those of WT but a faster activation kinetics and a voltage dependency shifted to more negative values (Fig. 4k-m, Supplementary Fig. 13a-d). R200Q mutant also showed a similar tendency similar to WT (Supplementary Fig. 13f-i). The charge neutralization in these mutants presumably decreases the interaction with anions, rendering the anions requiring more energy for transit. Consistent with this interpretation, R200N exhibits more negative membrane potential activation⁴⁹. R200 is located on the pore helices, so its side chain length is expected to have less impact on the interaction with the anions than R143. Notably, unlike R143 and R200, R226K/Q mutants were unable to conduct malate currents (Fig. 4k, m, Supplementary Fig. 13e). In the simulations, the application of a cytosolic negative membrane potential causes a R226 movement toward the cytosolic side (Supplementary Fig. 12d, e). Since all tested R226 substitutions disrupt ion conduction, both charge and shape of R226 seem to be crucial for AtALMT9 functionality.”

Authors write: Taken together, our results from MD simulations and vacuolar patch clamp recordings corroborate the structural roles of R143, R203 and R226 in activation of ALMT9.

Comment: Given that beside T143A all other mutations altered the properties of ALMT9 only marginally, the authors may want to tone down this statement.

Response: Pursuant to the reviewer’s suggestion, we revised the manuscript as follows. We would like to point out that in the revised manuscript, we now introduce the R226Q and R226K mutants exhibiting a strong effect on AtALMT9 function.

[Lines 417-478 of the revised manuscript]

“Taken together, the results from MD simulations and vacuolar patch clamp recordings corroborate the importance of R143, R200 and R226 in ALMT9 activation.”

Authors chapter: Conservation of key structural determinants in ALMTs

Comment: I read this rather long chapter several times but did not find the take-home message. When there is no story to tell the reader, this chapter should be condensed to a paragraph, like that in the Discussion: “Interestingly, sequence alignment suggest that this stepping hydrophilic pathway is conserved only in the clade 2 ALMTs (Fig. 6). These results could suggest that plugging/unplugging the pore from lipids may be involved in the regulation of ALMT9 ion transport activity and possibly in other ALMTs of the clade 2.”

Response: Pursuant to the reviewer's comment, we retained only the parts related to fenestration conservation for clarity and moved the remaining content to the “Description on Supplementary Fig. 16” in the Supplementary Information.

Authors write in the DISCUSSION, starting from: The clade 2 ALMTs constitute ...

Comment: the first paragraph is redundant and can be omitted.

Response: Pursuant to the reviewer's comment, the first paragraph is omitted.

Authors write: The clade 2 ALMTs constitute important anion channels responsible for transporting malate and chloride across the vacuolar membrane of guard cells which is essential for the regulation of stomata aperture.

Comment: In the Introduction and Discussion guard cells and the regulation of stomata aperture are mentioned. However, there is not a single result, that is gained with vacuoles from stomatal guard cells. There is also no experiment showing stomatal movement of guard cells expressing any of the mutants used in this study. Thus, mentioning open and closed stomata in the cartoon Figure 7 seems not justified.

Response: As the reviewer points out, we do not show stomata experiments. Therefore, we modified Fig. 7, which has become new Fig. 6 in the revised manuscript, such that we removed the words representing the stomata.

Question: Given that Kovermann et al. 2007 showed that ALMT9 is predominately expressed in mesophyll cell, the reader in the Introduction and/or Discussion would like to learn what the physiological role of malate induced vacuolar chloride fluxes are. Do we know about stimulus induced transients in cytosolic malate and associated chloride fluxes across the vacuole membrane?

Response: The study that the reviewer mentioned (Kovermann et al. 2007) has shown, among other tissues, the expression in the mesophyll. However, subsequently it was shown that AtALMT9 has major functions in the regulation of stomata opening (De Angeli, A., et al. 2013; Xu, B., et al. 2021). In guard cells, the malate concentration changes upon opening stimuli (e.g., light, fusicoccin) following starch degradation (Horrer, D., et al. 2016) and chloride is accumulated in the vacuole (Roelfsema and Hedrich. 2005). In this context, cytosolic malate level most likely changes and such a change could induce AtALMT9 activation.

Reviewer #2

ALMTs form a family of anion channel in plants and play important roles in the regulation plant anion homeostasis. Previous studies on AtALMT1 and GmALMT12 reveal overall architectures, anion selectivity, and gating mechanisms of ALMTs. In this study, the authors present structures of another ALMT protein AtALMT9 in different conformations, along with electrophysiological assays and computational studies. This manuscript is lengthy and inconclusive, and should be substantially revised before publication.

1. Although the overall map quality is good, the identification of lipids, malate, sterols, and cis1-PI4P solely based on the ligand density is questionable. For example, where did this cis1-PI4P come from? How did the authors distinguish it from POPC? Without more evidence, the authors cannot accurately assign these ligands, especially for malate.

Response: We agree that additional evidence is necessary to strengthen the claims about lipids. To prove the presence of lipids, we extracted hydrophobic molecules from AtALMT9 sample with the dataset 3 condition and we analyzed the extracts using liquid chromatography with tandem mass spectrometry (LC-MS/MS) (Fig. 1g-i, Supplementary Fig. 8). In lines with cryo-EM observations, we identified a lot of phospholipid with similar tail lengths from extracts (Supplementary Fig. 8). We revised the manuscript as follows to reflect this point:

[Lines 137-148 of the revised manuscript]

“To confirm the presence of PLs, we extracted lipids from the sample purified with the dataset 3 condition based on the Folch’s method⁴⁴. The lipid extract was analyzed by liquid chromatography with tandem mass spectrometry (LC-MS/MS). We detected various AtALMT9-bound PLs in the following descending order of abundance: phosphatidylethanolamine (PE), phosphatidylserine (PS), phosphatidylcholine (PC), phosphatidylinositol (PI), and phosphatidylglycerol (PG), but no detectable phosphatidic acid (PA) (Fig. 1g-i, Supplementary Fig. 8). In comparison with reported lipid compositions of Sf9 insect cells⁴⁵, anionic PS or acyl chain with 18 carbon atoms were slightly favored for AtALMT9 binding, but there was no significant preference for a specific lipid (Fig. 1g, h). These data were consistent with our structural observations showing lipid chains with 16 to 18 carbon atoms (Fig. 1h, Supplementary Fig. 7, 8).”

We detected the presence of phosphatidylinositol (PI), but not phosphatidylinositol phosphates (PIPs) likely due to suboptimal liquid chromatography solvents for recovering negatively charged PIPs. The lipid compositions observed by mass spectrometry and the electron densities of all classes suggest that PI should contribute to the cis1 class, not to other classes. We could not exclude that density might have been derived from PIPs or other PI derivatives. Accordingly, we changed the name of the cis1-PI4P class to cis1-PI/PIP and revised the manuscript as follows to reflect this point:

[Lines 160-167 of the revised manuscript]

“In the classes “cis1-PI/PIP” and “cis2”, two tails of the pore lipid are in the fenestrations of one protomer. In the class “intermediate”, only one tail of the pore lipid is in the fenestration. In cis1-PI/PIP class, the pore lipid was modeled as phosphatidylinositol 4-phosphate (PI4P) solely based on electron density. We observed scant densities for polar head groups of pore lipids while a bulky inositol head group adopted a relatively fixed position in the cis1-PI/PIP class (Supplementary Fig. 7). Mass spectrometry results suggest that such weak densities

stem from various head groups, but not PA with the phosphate group only (Supplementary Fig. 8).”

The abolition or decrease of malate flux in the R143, R200, and R226 mutants observed by electrophysiology experiments suggest malate binding by these residues. In simulations without membrane potential, we observed the prevalence of anion binding for short times at S1 and S3 sites in unplugged states. Consistently, a recent study reported a low affinity of AtALMT9 toward malate ($K_D = 2.12$ mM) and low abundance of malate binding by mass spectrometry (Qian, D., et al. 2024). We revised the manuscript as follows to reflect this point:

[Lines 366-374 of the revised manuscript]

“In the simulations at zero membrane potential, no anion flux was observed both in the N and W classes (Supplementary Fig. 12a-c, Supplementary Movie 1-3). Chloride or malate was found preferably at S1 and S3 in the unplugged classes, as in the cryo-EM structures (Fig. 4a, b, Supplementary Fig. 12a, b). The N class with chloride ion ($[Cl^-]^{High}$) showed negligible fluctuations in positions of the pore-lining arginine residues (R143, R200, and R226) at zero membrane potential (Supplementary Fig. 12a, d). By contrast, the W class with chloride ion and three malate molecules ($[Cl^-]^{High} + [malate]^{Low}$) exhibited dissociation of shallowly-inserted peripheral lipid tails from fenestrations. Consequently, R143 residues showed downward movements due to the absence of tails beneath it (Supplementary Fig. 12b, e).”

2. The sterol mimic-bound and lipid-bound AtALMT9 structures were overinterpreted. Clearly, these structures are in non-conducting conformations, which were probably induced during the protein sample preparation. To me, these structures, obtained from insect cells but used for the analysis of lipid-protein interactions for a plant ion channel, provide limited information for the channel activation. The authors should focus on the other two conformations.

Response: We found that the lipids detected in AtALMT9 obtained from insect cells were also present in the plant vacuolar membrane. Subsequently, the lipid binding states observed in this study are likely to occur in the plant vacuole. However, we cannot exclude a possibility that plant-specific lipids may contribute to the regulation of AtALMT9. We revised the manuscript as follows to reflect this point:

[Lines 547-550 of the revised manuscript]

“In mass spectrometry, we detected the existence of zwitterionic, anionic lipids, and PIs (Supplementary Fig. 8). These lipids are also present in the vacuolar membrane⁵⁸⁻⁶⁰ and the PIs have been reported to regulate stomata closure⁶⁰.”

[Lines 600-603 of the revised manuscript]

“Despite of extensive results, our study has certain limitations. The membrane of the heterologous expression system⁴⁵ differs from plant vacuoles⁵⁸⁻⁶⁰, thereby raising a possibility that some plant-specific lipids may have been ignored. Lipid preference or specificity of each plugged class is indeterminate due to the ambiguity of density, except the cis1-PI/PIP class.”

One may concern that addition of a detergent during purification of a membrane protein may introduce artifacts. By coincidence, the previously reported GmALMT12/QUAC1 (Qin, L., et al. 2022) and AtALMT9 in this study were purified with the same detergent, LMNG, and represented fenestrations. To investigate if there is any artifact in the structure of an ALMT due to a detergent, we determined the AtALMT1 structure using the same purification method employed in our ALMT9 study (Supplementary Fig. 21a-e). Specifically, we used LMNG as a detergent for purification of both AtALMT9 and AtALMT1 while the previously reported AtALMT1 was purified with a different detergent, GDN (Wang, J., et al. 2022). We found no significant difference between AtALMT1 structure and the previously reported one (Supplementary Fig. 21f). Interestingly, AtALMT9 structure in our study feature fenestrations while both the previously reported AtALMT1 structure and one in our study displayed no fenestration. Taken together, the sample preparation protocol for AtALMT9 seems to have minimal impact on conformation or lipid-bound states of the ALMT family. We revised the manuscript as follows to reflect this point:

[Lines 604-611 of the revised manuscript]

“Addition of a detergent LMNG may have introduced artifacts in protein purification. Notably, the purification of GmALMT12/QUAC1 whose structure has fenestration employed LMNG for purification⁸. To check whether lateral fenestrations are artifacts from our purification protocol, we determined the cryo-EM structure of AtALMT1 with the same purification method as with our AtALMT9 (Supplementary Fig. 21a-e). We observed no significant differences between the previously reported AtALMT1 structure with GDN (PDB: 7VQ4)²¹ and our structure with LMNG and CHS (Supplementary Fig. 21f). These results suggest that our detergent system has little impact on observed structures, especially the fenestration region.”

Reviewer #3

In the manuscript by Lee et al., the authors have resolved the structure of AtALMT9 in various lipid-bound states. Their study reveals that lipid tail length regulates the pore width, thereby allowing anion permeation. This research is of broad interest to the community working on channel proteins, particularly in understanding how lipids regulate gating mechanisms. However, I have a few concerns regarding the molecular dynamics (MD) simulations that the authors should address.

Major Comments:

1) Pressure Coupling in MD Simulations: The authors have turned off pressure coupling during their MD simulations. This choice can lead to unrealistic system behavior, especially in terms of membrane properties, protein-lipid interactions, and the overall physical realism of the simulation. Lipid dynamics within the bilayer, including flip-flop and lateral diffusion, are pressure-dependent. Turning off pressure coupling can result in non-physiological lipid behavior, which in turn affects membrane dynamics and the function of embedded proteins. Given that protein-lipid interactions are critical in this study, as they influence the pore widths, it is concerning that pressure coupling was not employed. Notably, several MD simulations studying lipid pore formation under external electric fields, even up to 600 mV, have been successfully conducted with pressure coupling turned on, yielding stable systems across various lipid types and electric fields (Reference: <https://doi.org/10.1021/jacs.2c08543>). I strongly recommend that the authors consider repeating their simulations with pressure coupling enabled.

Response: Following the review's suggestion, we repeated the all-atom MD simulations with semi-isotropic pressure coupling corresponding to zero surface tension. Additionally, we extended the simulation times from 1 microsecond to 3 microseconds. In these new simulations with pressure coupling, the box dimensions remained close to the initial values, indicating that the initial conditions were reasonable and the pressure coupling did not affect the lipid density in the presence of a potential difference. Compared to the previous simulations, the new simulations using pressure coupling are more consistent with the channel properties reported in a previous electrophysiological study (De Angeli, A., et al, 2013) such as single channel conductivity of conductive state, malate-dependent frequency increase of conductive phase and continuous flow of malate. We revised the manuscript as follows to reflect this point:

[Lines 376-394 of the revised manuscript]

“In the simulations at a cytosolic negative membrane potential, we observed that conformations of both N and W classes in the basal state converged into those in the conductive states representing both R143^{down} and R226^{up} (Fig. 4f-h, Supplementary Fig. 12d, e, Supplementary Movie 4-9). Such opposing movements of R143 and R226 rotamers allowed a new anion binding site (S2) between S1 and S3 by relieving hydrophobic constrictions of the pore in the basal state (Fig. 4f-h). At high malate concentrations ($[\text{malate}]^{\text{High}}$), three malate molecules occupied S1, S2, and S3. An entering malate pushed a pre-existing malate into the next site and a malate at S3 exited (Fig. 4f). Such ion transport is reminiscent of a knock-on mechanism in potassium channels⁴⁸. Such knock-on effects induced a continuous flow of malate ions. By contrast, chloride ions represented a sporadic influx due to chloride ion trapping among R143, R200, and R226 at S1 and S2 at high chloride concentrations ($[\text{Cl}^-]^{\text{High}}$) (Fig. 4g). In presence of a small portion of malate ($[\text{Cl}^-]^{\text{High}} + [\text{malate}]^{\text{Low}}$), a malate entering S2 disrupted chloride retention in S1 and S2,

thereby allowing the subsequent passage of many chloride ions (Fig. 4h-j). When comparing simulations ($[\text{Cl}^-]^{\text{High}}$ vs. $[\text{Cl}^-]^{\text{High}} + [\text{malate}]^{\text{Low}}$), the low malate concentration only increased the frequency of the conductive phase (Fig. 4g, h). In each simulation, both average transit times (20 - 26 ns) and average flow rates (142 - 177 ions per μs) of chloride ions were similar during conductive phase, regardless of malate ions (Fig. 4g, h). These simulation results are consistent with the malate-dependent open probability increase without conductivity changes in single-channel analysis⁹.”

[Lines 526-543 of the revised manuscript]

“Our MD simulations data shed light on the voltage-dependent functionality of AtALMT9. The cryo-EM structures of AtALMT9 most likely represent basal states at zero transmembrane potential. MD simulations applying a transmembrane potential revealed that the movements of side chains of R143 and R226 along with the removal of peripheral lipids contribute to the ion permeation (Fig. 4). Our MD simulation results are in line with experimentally observed properties of AtALMT9. Since 1 pA is equivalent to the transit of 6.24×10^6 monovalent ions per sec⁵⁷, we calculated that the chloride ion flow rates (142 - 177 ions per μs) in our simulations correspond to -22.8 to -28.4 pA at -500 mV (Fig. 4f-h). Single channel currents deduced from the MD simulations are in the range of -3.64 to -4.54 pA at -80 mV, consistent with the experimentally measured single channel currents of -2.7 to -2.8 pA at -80 mV⁹. Movement of R143 and R226 side chains in the ion permeation in conjunction with AtALMT9 functionality is corroborated by patch-clamp analysis using R143 and R226 mutants. While R143E, R143N, and all R226 mutants disrupt AtALMT9 functionality⁴⁹, we found that R143K and R143Q do not disrupt the channel functionality (Fig. 4k-m). It is noteworthy that R143Q modifies the voltage dependency of AtALMT9 and activation kinetics (Fig. 4l, Supplementary Fig. 13a-d). Taken together, the data obtained by the MD simulations show ion transport through the pore and provide significant insight into the voltage-dependent changes occurring in AtALMT9.”

2) Lipid Density Map Analysis: In lines 393-395, the authors mention that they set up MD simulations by replacing peripheral lipids modeled based on cryo-EM structures with POPC lipids. However, they have not provided a lipid density map analysis to confirm whether these lipids continue to occupy the cryo-EM observed binding sites during the MD simulations. This analysis is crucial to ensure the accuracy of the modeled interactions.

Response: Pursuant to the reviewer’s comment, we depicted lipids in fenestration as spheres and traces as transparent surface representations in all-atom simulations movies (Supplementary Movie 1-3). During the MD simulations we observed asymmetrical peripheral lipid bindings in both the narrow and wide structures, consistent with cryo-EM observations. Anion distributions are focused on S1 and S3 sites similar to cryo-EM. However, tail insertion depths and dynamic association/dissociation of peripheral lipids differ from cryo-EM structures. These inconsistencies between cryo-EM structures and MD simulations may have been caused by different environments or samplings as mentioned below. Nevertheless, the overall trends are similar in that peripheral lipid tails can bind to fenestrations.

From cryo-EM structure analysis, we obtained snapshots of conformational ensembles representing metastable lipid-binding states, indicating that each state is interchangeable to others. We processed only a limited population (55~56%) of particles, and the classified particles may contain a certain degree of heterogeneity. Considering that cooling effects in cryo-EM are attributed to the biased and narrow sampling of conformational landscape (Bock, L. V., et al., 2024), the true conformational ensembles are likely to be more diverse and

dynamic than cryo-EM observations. We also perceive those potential differences in membrane and micelle environments. In all-atom simulations at zero membrane potential, we were unable to obtain sufficient sampling of lipid movements around the fenestration and observed certain discrepancies in simulation and cryo-EM.

[Lines 366-374 of the revised manuscript]

“In the simulations at zero membrane potential, no anion flux was observed both in the N and W classes (Supplementary Fig. 12a-c, Supplementary Movie 1-3). Chloride or malate was found preferably at S1 and S3 in the unplugged classes, as in the cryo-EM structures (Fig. 4a, b, Supplementary Fig. 12a, b). The N class with chloride ion ($[\text{Cl}^-]^{\text{High}}$) showed negligible fluctuations in positions of the pore-lining arginine residues (R143, R200, and R226) at zero membrane potential (Supplementary Fig. 12a, d). By contrast, the W class with chloride ion and three malate molecules ($[\text{Cl}^-]^{\text{High}} + [\text{malate}]^{\text{Low}}$) exhibited dissociation of shallowly-inserted peripheral lipid tails from fenestrations. Consequently, R143 residues showed downward movements due to the absence of tails beneath it (Supplementary Fig. 12b, e).”

3) Lipid Tail Insertion and Stability: In lines 82-84, the authors discuss how the degree of lipid tail insertion from peripheral lipids regulates pore widths. Although they suggest that the tail lengths are likely 16 or 18 carbon atoms based on insect cell lipid composition (lines 129-133), they also acknowledge a small percentage of lipids with 20 carbon atoms. Longer lipid tails and unsaturation provide the structural flexibility needed to bind specific amino acids with their headgroups while positioning hydrophobic tails in the voids between helices. Given the flexible nature of lipid tails, it is challenging to confirm from the tube-like densities observed in cryo-EM whether they correspond to 16 or 18 carbon atoms. Moreover, the structural arrangement of lipid tails (C16/C18/C8/C10) shown in Figure 2 raises questions about their energetic stability. I suggest the authors conduct coarse-grained simulations with varying membrane compositions in tail length and degree of unsaturation. These simulations are relatively inexpensive and could help clarify which lipid tail configurations are energetically stable and best fit the tube-like densities observed in the cryo-EM structure.

Response: Pursuant to the reviewer’s suggestion, we conducted coarse-grained simulations for exploring lipid binding states in different membrane compositions for up to one hundred microseconds (Fig. 5i, Supplementary Fig. 19, Supplementary Movie 9-12). As expected, lipids exhibited dynamic binding throughout the coarse-grained simulations. We observed lipid migrations toward fenestration regions through hydrophilic membrane-facing side of the TM3 in the unplugged states, as well as lipid plugging in a similar manner in the plugged cis2 class. Intriguingly, different membrane compositions directly modulate lipid migration. Specifically, thick and dense hydrophobic layers formed by long, saturated tails hindered lipid movement across the hydrophilic surface, while thin and loose hydrophobic layers facilitated lipid movement. We also observed insertion of hydrophobic tails into the fenestration regions. However, it differs from the exclusive hydrophobic tail binding in the cryo-EM structures. This may have resulted from the distinct properties of hydrophobic layers in micelles and membranes, which can influence lipid movement, as observed in our simulations. We presumed that tail densities originated from the traces of various lipid tails or LMNG (with a length of 10 carbon atoms). Therefore, we modeled various pore and peripheral lipid tails, ranging from C8 to C18, based solely on their density shape. We verified the prevalence of C16 and C18 tails, with minor populations of C14, C20, and C22 tails by mass spectrometry (Supplementary Fig. 8). Hence, all observed densities may have resulted from the C16 and C18

tails (more than 97 % of the population), or from a portion of the LMNG responsible for the short density. We could not exclude the possibility that the head group partially contributes to the densities with short tails. However, long densities are unlikely to correspond to head groups due to their lengths. Nevertheless, these minor discrepancies do not pose significant issues in claiming lipid movements and plugging. It is worth investigating the details of lipid movements in future studies. We revised the manuscript as follows to reflect this point:

[Lines 488-506 of the revised manuscript]

“To corroborate the lipid migration through hydrophilic fenestration at an extended time scale, we conducted coarse-grained simulations using the Martini3 force field⁵⁴ for dozens of microseconds with the TMD structures in the narrow, wide, and cis2 classes (Fig. 5i, Supplementary Fig. 19, Supplementary Table 3, Supplementary Movie 10-12). We observed that hydrophilic moieties of lipids intruded the hydrophobic layer of the membrane along hydrophilic TM3, but not near the TM6 (Fig. 5i). Both the hydrophilic moieties and hydrophobic tails entered the fenestration in coarse-grained simulations, regardless of the TMD conformation (Fig. 5i). Unlike restricted pore access of lipids in unplugged states, the plugged cis2 class showed high occupancy of the pore lipids in the pore regions, as observed in the cryo-EM structure (Supplementary Fig. 19a, b). We also performed simulations with different membrane compositions according to tail lengths and saturation degrees (Supplementary Fig. 19c, d). The membrane thickness and stiffness seem to affect lipid migration toward the fenestration. The hydrophobic layer disruption of the membrane was unlikely to occur in the lipid bilayer of long saturated lipids owing to the dense and thick hydrophobic layer (Supplementary Fig. 19c, d). Contrary to the cryo-EM structures, we observed high occupancy of the hydrophilic moieties in the fenestration in simulations. It may come from different hydrophobic layer properties of lipid bilayer and micelle. Taken together, these results suggest that the hydrophilic lateral fenestration is likely to mediate the lipid migration for channel modulation.”

4) Fatty Acid Orientation in Figure 2a: In Figure 2a, the fatty acid tail is oriented such that the carboxyl group faces the hydrophobic bulk lipids, while the hydrophobic tail faces the pore. Have the authors considered rotating the fatty acid tail so that the carboxyl group faces the pore, potentially forming water-mediated electrostatic interactions with R143^{up}?

5) Fatty Acid Orientation in Figure 2b: Similarly, in Figure 2b, the peripheral fatty acids F_{A1}, F_{A2}, F_{B1}, and F_{B2} have their carboxyl groups facing the hydrophobic bulk lipids. Rotating the lipids would position the headgroups towards the pore, with the two acyl chains occupying the two tube-like densities. Have the authors considered this alternative lipid orientation, and would it remain stable at the binding site?

Combined responses to 4) and 5): We modeled that kind of density (peripheral lipid) as a fatty acid tail based on the assumption that the tail may have been derived from phospholipids on the membrane side. We already confirmed the tail-end density of all peripheral lipids (Fig. 2a, b, 3a), which were adopted on the hydrophobic interior of the pore. Sometimes, the hydrophobic ends of tails (F_{B1} of Fig. 2a, F_{A1} and F_{A2} of Fig. 2b) are weaker in density than other regions so we presumed that these properties originate from different tail lengths or insertion depths of phospholipids. If we ignore density weakness, the oppositely oriented fatty acid tail may be possible in the F_{B1} of Fig. 2a and form water-mediated electrostatic interactions with R143^{up}. However, no fatty acid was detected in mass spectrometry. It is worth exploring the effects of phospholipids and fatty acids on the activity of ALMT9 in future studies.

Minor Comments:

1) MD Simulation Methods Section: The MD simulation methods section lacks sufficient detail for reproducibility. For instance, the authors refer to results using structures representing unplugged states (Fig 5a, e and SI figures and movies), but the specific unplugged structures used in the MD simulations are not clearly referenced (e.g., Fig 2a, b). The authors should clarify the use of unplugged (N and W states) structures in the MD methods section and mention them in the SI Table 2 heading.

Response: Pursuant to the reviewer's comment, we revised the Results and Methods section to specify the models that were used in MD simulations.

[Lines 356-360 of the revised manuscript]

“To probe conductive states in AtALMT9, we performed molecular dynamics (MD) simulations using the unplugged narrow and wide structures (Fig. 2a, b, Fig. 4f-h, Supplementary Fig. 12a-c, Supplementary Table 2, Supplementary Movie 1-9). The TMD (residues 77-248) region in the narrow or wide class was inserted into the 1-palmitoyl-2-oleoyl-*sn*-glycero-3-phosphocholine (POPC) bilayer membrane after peripheral lipids were replaced by POPC.”

[Lines 843-844 of the revised manuscript]

“The AtALMT9 TMD (77-248) in the narrow or wide classes was embedded into a bilayer of POPC using the CHARMM-GUI webserver⁷⁸.”

2) Protonation States and Thermostat Parameters: The Methods section does not explain how the protonation states of the amino acids were determined, nor does it provide information on the total charge of the protein in different states or the input parameters used for the thermostat.

Response: We used a Gromacs tool (gmx pdb2gmx) to determine the protonation states based on the arrangement of side chains. However, determination of protonation states is not critical for this system, as the channel includes no histidine residues. Additionally, we clarified the input parameters for the thermostat and barostat in the Methods section.

[Lines 839-841 of the revised manuscript]

“To set the temperature at 300 K during simulations, we employed the v-rescale scheme. To set the pressure at 1 bar along the lateral and normal directions independently (i.e., zero surface tension), we used the c-rescale scheme with a semi-isotropic coupling option.”

3) Terminology Clarification in Lines 89-90: The authors could improve clarity by explicitly referring to the "peripheral lipid-only class" as "unplugged states."

Response: Pursuant to the reviewer's suggestion, we revised the manuscript accordingly and similar changes were applied to other sentences.

[Lines 87-88 of the revised manuscript]

“and two unplugged classes in a narrow and a wide pore.”

4) Movie Clarity: The movies provided by the authors are very clear. However, it would be beneficial to also include the peripheral lipids in the visualizations.

Response: Pursuant to the reviewer's comment, we revised the movies to include the peripheral lipids.

5) Data Availability: A link to a public repository containing the simulated structures and topology files is missing. Additionally, structures with PIP and cholesterol bound are absent. Uploading these structures to a public repository would significantly benefit the research community.

Response: We have uploaded simulation input files (including topology, initial structure, and Gromacs parameter file) as well as the trajectory files from the production runs to Zenodo.org (DOI: doi.org/10.5281/zenodo.14177701). The uploaded files are named consistently with the Supplementary Table 2 and 3.

References

1. Kovermann, P. *et al.* The Arabidopsis vacuolar malate channel is a member of the ALMT family. *The Plant Journal* **52**, 1169-1180 (2007).
2. Doireau, R. *et al.* AtALMT5 mediates vacuolar fumarate import and regulates the malate/fumarate balance in Arabidopsis. *New Phytologist* **244**, 811-824 (2024).
3. Jensen, M. Ø. *et al.* Mechanism of Voltage Gating in Potassium Channels. *Science* **336**, 229-233 (2012).
4. Kopec, W. *et al.* Direct knock-on of desolvated ions governs strict ion selectivity in K⁺ channels. *Nature Chemistry* **10**, 813-820 (2018).
5. Kasimova, M. A. *et al.* Helix breaking transition in the S4 of HCN channel is critical for hyperpolarization-dependent gating. *eLife* **8**, e53400 (2019).
6. Liu, C., Xue, L. & Song, C. Calcium binding and permeation in TRPV channels: Insights from molecular dynamics simulations. *Journal of General Physiology* **155**, e202213261 (2023).
7. De Angeli, A., Zhang, J., Meyer, S. & Martinoia, E. AtALMT9 is a malate-activated vacuolar chloride channel required for stomatal opening in Arabidopsis. *Nature Communications* **4**, 1804 (2013).
8. Xu, B. *et al.* GABA signalling modulates stomatal opening to enhance plant water use efficiency and drought resilience. *Nature Communications* **12**, 1952 (2021).
9. Horrér, D. *et al.* Blue Light Induces a Distinct Starch Degradation Pathway in Guard Cells for Stomatal Opening. *Current Biology* **26**, 362-370 (2016).
10. Roelfsema, M. R. G. & Hedrich, R. In the light of stomatal opening: new insights into 'the Watergate'. *New Phytologist* **167**, 665-691 (2005).
11. Qian, D. *et al.* Structural insight into the Arabidopsis vacuolar anion channel ALMT9 shows clade specificity. *Cell Reports* **43**, 114731 (2024).
12. Qin, L. *et al.* Cryo-EM structure and electrophysiological characterization of ALMT from *Glycine max* reveal a previously uncharacterized class of anion channels. *Science Advances* **8**, eabm3238 (2022).
13. Wang, J. *et al.* Structural basis of ALMT1-mediated aluminum resistance in Arabidopsis. *Cell Research* **32**, 89-98 (2022).
14. Bock, L. V., Igaev, M. & Grubmüller, H. Single-particle Cryo-EM and molecular dynamics simulations: A perfect match. *Current Opinion in Structural Biology* **86**, 102825 (2024).

POINT-BY-POINT RESPONSES

We sincerely appreciate the valuable comments and considerate decisions. Below we present our responses to the concerns raised by the reviewers.

Reviewer #1

Structural information on plant ion channel are quite rare and besides TPC1 vacuole ion channel 3D structures are even more so.

This state-of-art Cryo-EM study should be published.

The revised version of the paper has improved much concerning the story board and proper controls.

RESPONSE: We thank the reviewer for constructive comments in improving the depth of our study especially in electrophysiological aspects.

Reviewer #2

I am impressed by the authors' efforts in addressing my major concerns, as well as others'. In my opinion, this story presents too much experiment data from structures, EPs, to MD simulations. Although I do not totally agree with the authors' conclusion, given the other ALMT9 story, whose structures are in low resolutions, was published recently, in principle I would recommend the publication of this manuscript. Nevertheless, I wish the authors seriously consider my comments.

The structures presented in this study are convincing and cross-validated by the other study. However, the interpretation should be cautious. It is not unusual to observe lipids in the ion conducting pathway in ion channel structures, but not all these lipids are functional or physiology-relevant. To me, LMNG is not a perfect detergent and sometimes induces artifacts. As a control, the authors use LMNG to determine an ALMT1 structure. Why didn't they use a different detergent, for example, GDN, to determine the same ALMT9 as a control, which is more straightforward and relevant?

RESPONSE: We are grateful to the reviewer for thoughtful comments. We are fully aware that not all lipids observed in our structures may be fully functional. Further study will be needed to resolve these issues. We tested the solubilization of ALMT9 with other detergent such as GDN. A recent ALMT9 study (Qian, D., et al. 2024) presented the ALMT9 structures with digitonin or GDN micelles. To further corroborate no artificial effect by LMNG, we wished to confirm the effects of LMNG in other ALMT clades, particularly AtALMT1 in clade 1 which lacks fenestration. We revised the manuscript as follows to make these aspects explicit:

[Lines 614-621 of the revised manuscript]

“We observed no significant differences between the previously reported AtALMT1 structure with GDN (PDB: 7VQ4)²¹ and our AtALMT1 structure with LMNG and CHS (Supplementary Fig. 21f). Reported AtALMT9 structures with digitonin or GDN (PDB: 8HIW, 8HIY, 8ZVF)⁴⁶ also showed negligible differences with our AtALMT9 structures with LMNG (Supplementary Fig. 20). These results suggest that our detergent system has little impact on observed structures, especially the fenestration region. Nonetheless, some

questions remain regarding the physiological and functional relevance of lipids in channel regulation.”

Reviewer #3

After carefully reviewing the revised manuscript and the authors' responses to my previous comments, I am satisfied that all my concerns have been addressed. The authors have provided clear and thorough explanations, improving the overall quality and clarity of the manuscript.

I am pleased to recommend this manuscript for publication.

Minor Comment:

Authors could include a brief description in the Methods section explaining how chloride permeation events and lipid density maps were analyzed. This addition would enhance the reproducibility of their work for readers.

Overall, I commend the authors for their effort in addressing the feedback and presenting a well-prepared manuscript.

RESPONSE: We acknowledge the reviewer's insights in identifying potential issues in the simulations and providing valuable suggestions. To enhance reproducibility for readers, we added details on how to count ion permeation events and draw probability density for lipids in the Methods section. To validate reproducibility, we performed two additional independent simulations for non-equilibrium conditions (Supplementary Fig. 12f). We revised the manuscript as follows as suggested:

[Lines 380-383 of the revised manuscript]

“In the simulations at a cytosolic negative membrane potential, we observed that conformations of both N and W classes in the basal state converged into those in the conductive states representing both R143^{down} and R226^{up} (Fig. 4f-h, Supplementary Fig. 12d-f, Supplementary Movie 4-9).”

[Lines 417-419 of the revised manuscript]

“In the simulations, the application of a cytosolic negative membrane potential causes a R226 movement toward the cytosolic side (Supplementary Fig. 12d-f).”

Reference

1. Qian, D. *et al.* Structural insight into the Arabidopsis vacuolar anion channel ALMT9 shows clade specificity. *Cell Reports* **43**, 114731 (2024).